# Global Down-regulation of Gene Expression Induced by Mouse Mammary Tumor Virus (MMTV) in Normal Mammary Epithelial Cells

**DOI:** 10.3390/v15051110

**Published:** 2023-05-02

**Authors:** Waqar Ahmad, Neena G. Panicker, Shaima Akhlaq, Bushra Gull, Jasmin Baby, Thanumol A. Khader, Tahir A. Rizvi, Farah Mustafa

**Affiliations:** 1Department of Biochemistry & Molecular Biology, College of Medicine and Health Sciences (CMHS), United Arab Emirates (UAE) University, Al Ain 15551, United Arab Emirates; waqar.ahmad@uaeu.ac.ae (W.A.); 201990158@uaeu.ac.ae (N.G.P.); 201570084@uaeu.ac.ae (S.A.); 201790692@uaeu.ac.ae (B.G.); 201790693@uaeu.ac.ae (J.B.); 201890024@uaeu.ac.ae (T.A.K.); 2Department of Microbiology and Immunology, College of Medicine and Health Sciences (CMHS), UAE University, Al Ain 15551, United Arab Emirates; tarizvi@uaeu.ac.ae; 3Zayed Center for Health Sciences (ZCHS), UAE University, Al Ain 15551, United Arab Emirates; 4ASPIRE Research Institute in Precision Medicine, Abu Dhabi, UAE University, Al Ain 15551, United Arab Emirates

**Keywords:** mouse mammary tumor virus (MMTV), mammary epithelial cells, HC11 cells, mRNAseq, microRNAs, differential gene expression, breast cancer

## Abstract

Mouse mammary tumor virus (MMTV) is a *betaretrovirus* that causes breast cancer in mice. The mouse mammary epithelial cells are the most permissive cells for MMTV, expressing the highest levels of virus upon infection and being the ones later transformed by the virus due to repeated rounds of infection/superinfection and integration, leading eventually to mammary tumors. The aim of this study was to identify genes and molecular pathways dysregulated by MMTV expression in mammary epithelial cells. Towards this end, mRNAseq was performed on normal mouse mammary epithelial cells stably expressing MMTV, and expression of host genes was analyzed compared with cells in its absence. The identified differentially expressed genes (DEGs) were grouped on the basis of gene ontology and relevant molecular pathways. Bioinformatics analysis identified 12 hub genes, of which 4 were up-regulated (Angp2, Ccl2, Icam, and Myc) and 8 were down-regulated (Acta2, Cd34, Col1a1, Col1a2, Cxcl12, Eln, Igf1, and Itgam) upon MMTV expression. Further screening of these DEGs showed their involvement in many diseases, especially in breast cancer progression when compared with available data. Gene Set Enrichment Analysis (GSEA) identified 31 molecular pathways dysregulated upon MMTV expression, amongst which the PI3-AKT-mTOR was observed to be the central pathway down-regulated by MMTV. Many of the DEGs and 6 of the 12 hub genes identified in this study showed expression profile similar to that observed in the PyMT mouse model of breast cancer, especially during tumor progression. Interestingly, a global down-regulation of gene expression was observed, where nearly 74% of the DEGs in HC11 cells were repressed by MMTV expression, an observation similar to what was observed in the PyMT mouse model during tumor progression, from hyperplasia to adenoma to early and late carcinomas. Comparison of our results with the Wnt1 mouse model revealed further insights into how MMTV expression could lead to activation of the Wnt1 pathway independent of insertional mutagenesis. Thus, the key pathways, DEGs, and hub genes identified in this study can provide important clues to elucidate the molecular mechanisms involved in MMTV replication, escape from cellular anti-viral response, and potential to cause cell transformation. These data also validate the use of the MMTV-infected HC11 cells as an important model to study early transcriptional changes that could lead to mammary cell transformation.

## 1. Introduction

Mouse mammary tumor virus (MMTV) is a milk-transmitted retrovirus that belongs to the genus *betaretrovirus*. MMTV has been known to cause mammary tumors in mice since the early 19th century [1,2]. It is secreted in the mother’s milk by the mammary epithelial cells and targets dendritic, B-, and T-cells in the Peyer’s patches found in the gut of suckling pups [3]. The infected lymphocytes circulate within the animal and transport the virus to the mammary gland where hormonal stimulation during puberty and pregnancy leads to increased virus replication, infection/superinfection, and integration, leading to insertional activation of growth promoting genes and eventually their carcinogenesis [4,5,6]. MMTV infects the host cells after binding with the mouse transferrin receptor1 (mTfR1) using its envelope glycoprotein (Env) [7,8]. This interaction results in endocytosis of the virus into a late, low pH endosomal compartment, which allows fusion of the virus with the host cell membrane and release of the capsid into the cytoplasm [9]. The process of uncoating initiates reverse transcription of the genomic RNA into complementary DNA (cDNA), followed by its integration into the host genome, a process which occurs in the most random manner amongst all retroviruses [4,10]. MMTV can also be transmitted as an endogenous virus via the germline, and numerous strains of such endogenous viruses have been identified, referred to as *Mtvs* [11]. Most of the endogenous strains of *Mtvs* are observed to be defective, unable to cause mammary carcinogenesis; however, a few non-mutated, replication-competent *Mtvs* observed in inbred mice cause mammary tumors [3,12,13,14]. Whether exogenous or endogenous, different strains of MMTVs reveal more than 95% similarity over their genomes other than the region encoding *sag*, which is the most variable part of the viral genome [11,15,16].

It is well known that upon viral infection, the host triggers an innate immune response against the virus as a defensive mechanism. This includes recognition of pathogens by “pathogen-associated molecular patterns (PAMPs)”, such as those on double-stranded RNA via triggering pattern recognition receptors (PRRs), including Toll-like receptors (TLRs), Aim-2-like receptors, and cyclic GMP-AMP synthase (cGAMPs). The innate immune response also includes pro-inflammatory cytokines, autophagy, and production of type I/II interferons (IFN) by mediating NF-kB and/or JAK-STAT pathways [17,18,19,20,21,22]. Although these mechanisms are not fully understood, it is known that early reverse transcription activity of MMTV could be blocked either by stimulating PAMPs that sense the presence of invading pathogens and trigger type I immune responses or by the activation of the host apolipoprotein editing complex 3 (APOBEC3) [23,24]. In response, the pathogens also try to evade such host defensive measures to continue their replication by exploiting the host defense systems [25]. For instance, studies have revealed that MMTV requires the host gut microbiota for its transmission after manipulating the bacterial lipopolysaccharides (LPS) since LPS is recognized by TLRs that, in turn, activate immune-responsive cytokines [26,27,28]. That is why germ-free (GF) and antibiotic-treated mice remain free from MMTV infection for many generations [27]. Thus, successful virus infection, replication, and pathogenesis are intricately dependent upon interconnected virus-host interactions.

Upon infection, viruses can also inhibit apoptosis of host cells and dysregulate caspases by encoding homologs of the anti-apoptotic proteins, such as Bcl-2, blocking apoptosis signals by triggering expression of the tumor necrosis factor (TNF) family members, or by inactivating IFN-induced protein kinase R (PKR) or p53 pathways that result in tumor formation [29,30,31]. It is interesting to note that unlike other cancer-causing viruses, MMTV does not encode any oncogene, and MMTV-associated carcinogenesis depends upon the host immune response, viral strain, viral proteins, especially Env and Sag, host–immune response modification by the virus, or the presence of endogenous *Mtvs* [32,33,34,35]. Furthermore, MMTV may also be jumping the species barrier into humans since MMTV-like antigens, virus particles, and sequences have been detected in the human milk, both in normal and breast cancer samples [36,37,38,39,40]. Moreover, a cloned MMTV provirus has been shown to effectively infect mouse as well as human cells [41,42]. Indeed, if this is true, there is a need to comprehensively study how MMTV expression affects the host cellular environment for its benefit.

Despite significant research to understand MMTV–host interactions, considerable work is still required to fully understand the underlying host molecular mechanisms involved in pathogenesis and carcinogenesis after MMTV infection. In this study, we established a mouse mammary epithelial cell line (HC11) stably expressing MMTV and used high-throughput RNAseq to profile any changes in the mRNA levels that may contribute to different molecular pathways associated with MMTV expression. HC11 are ideal mammary epithelial cells frequently used for the study of MMTV replication, gene expression, and virus–host interactions [43,44,45]. The HC11 cell line is also considered an excellent in vitro model for prolactin-induced differentiation since these cells represent the natural characteristics of stem-like epithelial cells resident in the mammary gland [46,47]. During the normal life cycle of MMTV in the mouse, the mammary epithelial cells serve as the main targets of MMTV infection cells that eventually give rise to the virus-induced breast tumors as well [25]. The HC11 cell line was isolated from the mammary glands of a pregnant BALB/c mouse and, thus, has been used extensively to study not only normal cell functions, such as mammary cell proliferation, differentiation, and signaling pathways, but also mammary cell transformation and milk production [8,46]. Interestingly, gene expression in these cells in their stem-like state has been shown to be quite similar to that observed in human breast cancer cells. Comparable expression of several genes and pathways in HC11 and breast cancer cells suggests usefulness of the HC11 cell line in understanding the mechanism involved in breast cancer [48,49,50,51,52,53,54]. It also suggests that HC11 cells are a novel tumor model for testing anti-cancer and anti-inflammatory agents [55,56,57]. In addition to the identification of differentially regulated genes (DEGs), we conducted network analysis to understand interaction between DEGs that form biologically relevant networks perturbed by MMTV expression. Thus, these results should help to better understand how MMTV exploits the host molecular pathways to evade the innate immune responses, establish infection, and induce cell transformation.

## 2. Materials and Methods

### 2.1. MMTV-Expressing Stable Cell Line

The normal BALB/c mouse mammary epithelial cells, HC11, were used to study the effect of virus expression on HC11 gene expression. The HC11 cells used in this study were a gift from Prof. Jeffery M. Rosen, Baylor College of Medicine, Houston, TX, USA. The cells were maintained in complete growth medium containing RPMI-1640 (Hyclone, Logan, UT, USA), 10% fetal bovine serum (FBS; Hyclone, Logan, UT, USA), 5 ug/mL insulin (Sigma-Aldrich, St. Louis, MI, USA), 0.01 ug/mL epidermal growth factor (EGF; Sigma-Aldrich, St. Louis, MI, USA), and 1% penicillin–streptomycin. The molecular clone of MMTV, HYB MTV [58] was used to transfect HC11 cells using HD Fugene transfection kit according to the manufacturer instructions (Promega, Madison, WI, USA). Hygromycin-resistant colonies were selected for two weeks using 200 µg/mL hygromycin and screened for MMTV expression using Western blot analysis.

Protein lysates from HC11 and HC MMTV cells were prepared in RIPA buffer (10 mM Tris–HCl (pH 8.0), 1 mM EDTA, 140 mM NaCl, 0.1% sodium deoxycholate 1% Triton X-100, and 0.1% SDS) and quantified using Bradford assay (Bio-Rad Laboratories, Hercules, CA, USA). A total of 40 ug protein was used for each sample to check for MMTV expression on a 4–12% gradient SDS gel (GenScript ExpressPlus PAGE, Piscataway, NJ, USA) and transfer was carried out overnight at 30V at 4 °C. The membrane was blocked for 1 h at room temperature in 5% non-fat milk, followed by overnight incubation in rabbit polyclonal MMTV anti-Gag primary antibody (Rockland Immunochemicals Inc., Limerick, PA, USA, 100–401-P12, USA) at a dilution of 1:1000 in 2% non-fat dry milk. The membrane was incubated with the respective HRP-conjugated secondary antibody. The blot was developed with ECL Plus Western blotting substrate (Thermo Fisher Scientific, Waltham, MA, USA), and the resulting chemiluminescent signals were detected using Typhoon FLA 9500. A similar procedure was followed for testing β-Actin (1:25,000 dilution of the primary antibody (Sigma-Aldrich, catalogue no. A3854, St. Louis, MI, USA) in 2% non-fat dry milk.

### 2.2. RNA Extraction, mRNA Sequencing, and Gene Expression Validation Using Real-Time Quantitative PCR (RT-qPCR)

Total RNA was extracted from undifferentiated HC11 cells in the presence and absence of MMTV using the TRIzol Reagent (ThermoFisher Scientific, Waltham, MA, USA) and quantified using Nano Drop as described previously [59]. Whole-cell RNA from two independent passages of these cell lines was sequenced commercially by the Beijing Genomics Institute (BGI, Hong Kong) using TruSeq library and DNBSEQ platform. The RIN values for the RNA samples ranged between 8.5 and 9.5 as assessed by the Agilent 2100 Bioanalyzer. Before sequencing, total RNA was treated with DNase I to remove any contaminating DNA, and the mRNA was enriched using oligo dT magnetic beads. This was followed by cDNA synthesis, end repair, addition of A and adaptor ligation, PCR, circularization, and sequencing (https://www.bgi.com/wp-content/uploads/sites/4/2017/06/F17FTSAPJT0170_FUNxovR_report_en.pdf, accessed on 23 November 2021). The average yield was 9.98 G data per sample, whereas average alignment ratio of the sample comparison genome was 94.13%.

Expression profiles of ten randomly selected genes identified by RNAseq were verified by quantitative real time PCR (RT-qPCR). Towards this end, total RNA was converted into cDNA by reverse transcribing 6 µg DNase-treated total RNA by MMLV RT (Promega, Madison, WI, USA) according to manufacturer’s instructions, as described before [60]. The cDNAs prepared were subjected to qPCR using 5X HOT FIREPol EvaGreen^®^ qPCR Mix Plus (ROX) (Solis Biodyne, Tartu, Estonia) and the QuantStudio 7 Flex real time system from Applied Biosystems, ThermoFisher Scientific). KiCqStart^®^ pre-designed primers from Sigma, and some from other published studies were used according to manufacturer’s directions or conditions listed in Table 1. β-actin was used as the endogenous control. The PCR reactions were performed in either 96- or 384-well plates in duplicates containing the primer mix and 1 µL cDNA, making the total reaction volume to 20 µL (96-well plate) or 10 µL (384-well plate) with DNase-Rnase free molecular biology grade water. All primer pairs were initially tested to ensure that each pair gave a single band and a single peak upon melt curve analysis.

### 2.3. RNAseq Data Pre-Processing

RNAseq was performed on control and MMTV-expressing HC11 total RNA samples in duplicates, named CTRL_1_, CTRL_2_, MMTV_1_, and MMTV_2_, respectively. The raw data were filtered using BGI in-house software known as SOAPnuke v1.5.2 [61], and the reads of low quality, adaptor contamination, and excessively high levels of unknown base N were removed (N > 5%). The Pearson correlation coefficients were calculated to compare the quality of gene expression between each sample, which showed higher similarities between each sample in each group (Appendix A). Principal component analysis (PCA) was used to remove any outliers from samples. The filtered clean reads were next subjected to quality control using FASTQ [62].

Finally, a total of 102.42 million reads (100 bp length) were generated for each sample and subjected to quality control. Raw data were cleaned, which limited the read count to 101.6, 100.46, 102.45, and 102.45 million reads for CTRL_1_, CTRL_2_, MMTV_1_, and MMTV_2_ samples, respectively. The clean reads were aligned to the reference mouse genome sequence GCF_000001635.26_GRCm38.p6 (*Mus musculus*), with an average alignment of 94.13% using HISTAT2. The reference genes were aligned to the data using Bowtie2 [63]. The average alignment of the gene set was 81.98%, and a total of 17,346 transcripts were detected (see Appendix A). The results were submitted to the BGI in-house software Dr. Tom accessed through an online server for further analysis. The raw and analyzed data (BioProject accession number: PRJNA915407) can be downloaded from the server for data re-analysis and further processing.

### 2.4. Identification of DEGs

After sequencing and cleaning (adaptor removal), the raw data in fastq file(s) format were analyzed using automated Dr. Tom software from BGI that allowed visualization and analysis of raw data. Using this software, we retrieved 17,346 transcripts that were further analyzed for quality control. The clean reads were mapped to the reference genes using Bowtie2 v2.2.5, and RSEM v1.2.8 [64] was used to calculate gene expression level of each sample. The DEseq2 method [65] (Q-value/ adjusted *p*-value ≤ 0.05) was used to detect differentially expressed genes. The genes with adjusted *p*-value of ≤0.05 and log2FC ≥ 2/≤ −2 were considered as differentially expressed genes (DEGs) and presented in red (up-regulated) or green (down-regulated) throughout the manuscript. All of the data were downloaded from Dr. Tom and re-analyzed using Microsoft Excel for any discrepancies. Venn diagrams were drawn for the overlapping genes/transcripts using the online platform Bioinformatics & Evolutionary Genomics from Van de Peer lab website (http://bioinformatics.psb.ugent.be/webtools/Venn/, accessed on 19 February 2022). Heatmaps were generated using Multiple Experiment Viewer v4.9.0 [66]. The volcano and other plots used in this study were generated through Dr. Tom and were further improved accordingly.

### 2.5. Functional Enrichment of Gene Ontology (GO) and Pathway Analysis

The DEGs list was uploaded to DAVID (https://david.abcc.ncifcrf.gov/tools.jsp, accessed on 22 February 2022) [67] for gene ontology (GO) and pathway analysis using *Mus musculus* as the reference species. GO analysis plots genes according to their functions, biological processes, cellular presence, and molecular functions. DAVID generated the list of genes involved in several biological pathways using KEGG (Kyoto Encyclopedia of Genes and Genomes: https://www.genome.jp/kegg/pathway.html, accessed on 12 April 2022) [68] and Reactome (https://reactome.org/, accessed on 14 April 2022) [69] pathway databases. We also searched the Wikipathways (https://www.wikipathways.org/index.php/WikiPathways, accessed on 26 April 2022) [70] for pathways associated with the DEGs. All of these databases have limitations, and the DEGs whose GO or pathways were not defined using these databases were further searched using GeneCards (https://www.genecards.org/, accessed on 4 May 2022) [71] and Rat Genome Database for mouse species (RGD: https://rgd.mcw.edu/wg/species/mouse/, accessed on 11 May 2022) [72]. The genes were sorted on the basis of the combined data from all these sources and used for further analysis, unless otherwise stated. The genes without any verified information were shown as “uncharacterized”.

### 2.6. Gene Sets/Pathways Enrichment Analysis

Gene Set Enrichment Analysis (GSEA) v4.10 [73] was used to evaluate the possibility of gene set associations with specific phenotypes. GSEA was performed using expression sets of whole genes rather than DEGs. The gene sets were retrieved from KEGG database for individual pathways. GSEA run used the following parameters: 100 permutations, “weighted” as enrichment statistic and “Signal2Noise ratio” as metrics for ranking the genes. GSEA gives both significant and non-significant pathways on the basis of nominal p-values and FDR q-values. GSEA uses FDR q-value < 0.25 for further analysis (https://software.broadinstitute.org/cancer/software/gsea/wiki/index.php/FAQ#Why_does_GSEA_use_a_false_discovery_rate_.28FDR.29_of_0.25_rather_than_the_more_classic_0.05.3F, accessed on 22 June 2022). Due to this reason, some pathways with *p*-value > 0.05 were also selected for further analysis. Pathways were further filtered down by selecting the sets having family-wise error rate (FWER) *p*-value ≤ 0.2 (the probability of making one or more false discoveries).

### 2.7. Construction of the Gene–Pathway Network, Protein–Protein Interaction (PPI) Network, and Hub Gene Network

The significant pathways were interrogated for overlapping and unique DEGs representing individual pathways. The online database STRING v11.5 [74] was used to construct PPI networks, while Cytoscape v3.8.2 [75] was used to view these networks and for constructing gene–pathway interactions networks. CytoHubba [76], a plugin of Cytoscape, was used to detect the hub genes from network analysis, whereas MCODE [77] was used to identify any significant module for PPI networks.

### 2.8. Functional Analysis of Hub Genes with Current Data

The online Comparative Toxicogenomics Database (CTD: http://ctdbase.org/, accessed on 13 April 2022) [78] was used to further explore association of the hub genes with diseases of interest, whereas The Cancer Genome Atlas (TCGA: https://www.cancer.gov/about-nci/organization/ccg/research/structural-genomics/tcga, accessed on 16 April 2022) was used for searching expression profile of key hub genes in cancer patients.

### 2.9. Prediction of miRNAs

The possible miRNAs that may target the identified hub genes were predicted using miRDB database (http://mirdb.org/mirdb/index.html, accessed on 20 April 2022). All miRNAs were investigated using the tab “Search by gene target” using “Mouse” as the reference species.

## 3. Results

To determine the effect of MMTV expression on normal mouse mammary epithelial cell gene expression, we established a stable MMTV expressing cell line (MMTV) using a replication-competent molecular clone of MMTV, HYBMTV [58] in the normal mouse mammary epithelial HC11 cells, a spontaneously immortalized, non-tumorigenic cell clone from the COMMA-ID mammary epithelial cell line isolated from the BALB/c mice [8,46]. Mammary epithelial cells are the main targets of MMTV infection in the mouse mammary gland that ultimately give rise to the mammary tumors [25]. HC11 cells can be infected by MMTV, but rather inefficiently [60]; therefore, we chose to create stably transfected HC11 cells, where all our cells were expressing the virus. This increased our chances of detecting subtle changes in MMTV-induced gene expression effectively. Furthermore, MMTV does not show superinfection resistance [79]; therefore, these cells should still be undergoing natural multiple rounds of “infection” with the virus produced from the stable cells, creating an environment similar to the “naturally infected” mammary cell. Expression of the MMTV structural polyprotein Gag was confirmed in these cells using an anti-Gag polyclonal antibody (Figure 1a). Two independent passages (biological replicates) of the HC11 cells (named CTRL1 and CTRL2 in this study) and HC11 cell expressing MMTV (named MMTV1 and MMTV2) were used for mRNAseq analysis to determine how MMTV affected various gene expression pathways in these cells.

RNAseq itself is a robust technique and may not require any additional validation [80,81]. However, to further confirm our findings, we selected 10 genes from the RNAseq data belonging to different pathways and used RT-qPCR to check their expression in control and MMTV-infected samples. Our RT-qPCR data showed that except for one gene, the expression profile of the remaining nine genes was in accordance with the RNAseq data (Figure 1b). These results agree with previous findings where comparative analysis between RNAseq and RT-qPCR showed 15–20% non-concordance gene expression, which could be due to several factors, such as the level of expression, sensitivities of different assays, and efficiency differences between PCR primers [80,81].

### 3.1. Identification of DEGs in MMTV Infected Mammary Epithelial Cells

RNAseq analysis revealed that of the 17,346 transcripts retrieved, both the control (CTRL) and MMTV-expressing (MMTV) samples shared 15,619 transcripts, whereas 900 transcripts were unique to CTRL and 827 to MMTV (Figure 2a and Appendix A). Overall, the data quality was acceptable, as box plots exhibited the confidence of the data within each replicate in both groups with fairly consistent medians across all four samples (Figure 2b). As we used two biological replicates of the pooled stable cell line, we averaged as well as leveraged the control and experimental groups. Averaging and leveraging not only increase reliability but also the sensitivity of the data. It should be noted that in most of the sequencing data, quality is preferred over quantity [82]. Gene expression levels found in our study confirmed both consistency as well as reliability of the data. Expression levels of the DEGs were visualized by creating a heatmap of the differentially up- and down-regulated genes (Figure 2c).

Hierarchal clustering verified the equal data distribution within each sub-group, thus verifying data consistency further. The unique transcripts within each group were filtered out, and the remaining 15,619 common transcripts were mined for the differentially expressed genes (DEGs). Dr. Tom used DEseq2 method to identify any DEGs between two groups, and transcripts with fold change logFC ≥ 2, *p*-value < 0.05, and Q-value < 0.1 were identified as DEGs. Similar filters were used manually to verify the data. Scatter plot showed the normal distribution of read-counts and significant genes with satisfying values (logFC ≥ 2 and logFC ≤ 2, *p*-value < 0.05, and Q-value < 0.1) that were selected as DEGs (Figure 2d). These genes are shown as red or green dots for up- and down-regulated genes, respectively. A total of 965 genes were differentially regulated within both groups, of which 249 (26%) genes were up-regulated (red), while 716 (74%) were down-regulated, revealing a significant down-regulation of gene expression following MMTV expression (green; Figure 2e; see Appendix A for details). Finally, a volcano plot of all genes shows the distribution of expression in these differentially regulated profiles (Appendix A). As can be seen, nearly a third of the genes were observed to be highly down-regulated compared with the upregulated genes.

### 3.2. Gene Ontology (GO) Analysis of the DEGs

GO analysis including functional, biological, cellular, and molecular functions of the DEGs was performed either by Dr. Tom’s online module (https://biosys.bgi.com/, accessed on 10 December 2021) or manually using DAVID v6.8 (https://david.ncifcrf.gov/home.jsp, accessed 22 February 2022), an online server. For genes that were not listed in either of these databases, several other online databases were used (including GeneCards (https://www.genecards.org/, accessed on 4 May 2022), RGD (https://rgd.mcw.edu/wg/species/mouse/, accessed on 11 May 2022), and STRING (https://string-db.org/, accessed on 24 May 2022). For pathway annotation, KEGG (Kyoto Encyclopedia of Genes and Genomes) database (https://www.genome.jp/kegg/pathway.html, accessed on 12 April 2022) was used as a preliminary tool. The genes not listed in the KEGG were then searched for their pathway association by using WikiPathways (https://www.wikipathways.org/index.php/WikiPathways, accessed on 26 April 2022) and/or Reactome Pathway (https://reactome.org/, accessed on 14 April 2022) databases. Every gene was searched for its function, and the genes were placed in specific categories on the basis of the accumulative information from all sources described above. The genes without any verified information were shown as “uncharacterized”. Figure 3 shows the top ten annotations for each category based on the number of genes in each set, while the remaining can be found in Appendix A. Functional annotation analysis of the DEGs showed that most of the genes belonged to membrane, glycoproteins, signaling, and metal-binding categories. We found transport, signal transduction, organism development, transcriptional regulation, cellular adhesion, proteolysis and cell differentiation, and proliferation as major biological processes. Most of the DEGs were part of the cellular membrane, surface, and junctions, especially the extracellular regions. Molecular functional characterization of the DEGs showed their involvement in hydrolyzations, ion binding, and signal transduction and function as catalysts and GTPase activators (Figure 3a).

We also searched for the involvement of the genes in biological pathways. Figure 3b represents the accumulative information for every gene driven from different data sources, as described above. Most of the DEGs were overlapping among many pathways (see Appendix A). DAVID analysis found several signaling pathways associated with DEGs, including calcium, metabolism, immune response, apoptosis, cell cycle, cytokine-cytokine, PI3-Akt-mTOR, Egfr, Mapk, insulin, Wnt, focal adhesion, Tnf, glutathione, Notch, and p53 (Figure 3b).

On the basis of the presence of numerous DEGs in multiple pathways, we generated a list of up- and down-regulated genes that were present in at least five or more pathways. We found 122 such genes, where 27 (22%) were up-regulated (red bars) and 95 (78%) were down-regulated (green bars), maintaining the significant down-regulation of gene expression observed earlier upon MMTV expression (Figure 3c; Appendix A).

Initially DEGs identified in our study were grouped into 54 known biological pathways; however, only a total of 38 gene sets (biological pathways) retrieved from KEGG database were tested (Table 2; Appendix A). GSEA showed that 9 gene sets were enriched (up-regulated) upon MMTV expression (the MMTV group), whereas 29 were enriched in the CTRL group (down-regulated) upon MMTV expression; Figure 3). Further analysis showed that out of the nine pathways, six were significantly enriched upon MMTV expression (FDR q-values < 0.25), which was further narrowed down to three on the basis of FWER *p*-values (<0.2), including Tnf, autophagy, and type II interferon pathways. For the CTRL group, 25 pathways showed FDR q-values < 0.25. After withdrawing PI3-AKT (as it shares the same genes as the PI3-AKT-mTOR pathway), we chose 13 pathways with FWER *p*-value < 0.2 (Table 2) for further analysis. These included the Wnt signaling, hedgehog, focal adhesion, Rap1, metabolism, PI3-AKT-mTOR, prolactin, Egfr, Hippo signaling, Ras signaling, inflammation, glutathione, and Vegf pathways.

### 3.3. Gene Enrichment Analysis for the DEGs Associated with Different Pathways

Following the DEGs identification, GSEA was employed to identify any changes in the expression/trend of the biological pathways after MMTV expression. GSEA is a useful tool that points out enrichment of a particular gene dataset within specific phenotypes on the basis of change in overall expression of the dataset. Figure 4 shows the enrichment plots of the top 16 pathways selected for further investigation, whereas Table 3 contains the list of genes that were significantly enriched within each pathway (including both DEGs and non-DEGs). Detailed information about GSEA and results interpretation can be found on http://www.gsea-msigdb.org/gsea/doc/GSEAUserGuideFrame.html?_Interpreting_GSEA_Results [73]. Here, we briefly described the method and terms used in GSEA in our study. The enrichment score reflects the degree to which a gene set (or pathway) is over-represented at the top or bottom of a ranked list of the genes. GSEA calculates the enrichment score (ES) ES by walking down the ranked list of genes, increasing a running-sum statistic when a gene is in the gene set and decreasing it when it is not. The magnitude of the increment depends on the correlation of the gene with the phenotype. The ES value is the maximum deviation from zero encountered in walking the list. A positive ES indicates gene set enrichment at the top of the ranked list; a negative ES indicates gene set enrichment at the bottom of the ranked list. A positive value indicates correlation with the first phenotype, and a negative value indicates correlation with the second phenotype. Any enrichment plot can be divided into three parts: (i) the peak point of the green line represents the ES for a particular gene set, (ii) the middle of the portion (red–blue horizontal and gray vertical lines) of the graph shows where the member of a subset falls in the ranked list of genes, and (iii) the lower portion of the graph (gray) shows the distribution of the gene set. The normalized enrichment score (NES) has been used primarily for the enrichment results and accounts for any differences of gene sets among phenotypes. The NES could be retrieved by dividing the actual ES by mean ES. Gene sets showing false discovery rate (FDR) of <0.25 (or 25%) show that the result is likely to be valid three out of four times. Thus, we selected the gene sets that had FWER *p*-value < 0.2 to ensure significance and further reduce the number of data sets. However, it is important to keep in mind that this does not mean that other datasets with FDR < 0.25 and FWER > 0.2 are not significant. As GSEA goal is to engender a hypothesis, it is recommended to focus more on FDR rather than FWER.

### 3.4. Determination of the Interaction among Core Genes and Biological Pathways Involved upon MMTV Expression

The top 16 significant pathways in both MMTV (n = 3) and CTRL (n = 13) phenotypes depicted in Table 3 were further analyzed for the member genes within each pathway and their interactions with other pathways. To do so, we constructed a pathway–gene interaction (PGI) network using Cytoscape v3.8.2. As GSEA uses whole genes in the pathway, either DEGs or non-DEGs, we removed the non-DEGs from the enriched genes sets. The constructed PGI network was composed of 211 nodes (genes) and 258 edges (Figure 5; Appendix A). The PGI network showed that many of the genes were shared by multiple pathways, which confirmed our initial analysis that these genes could be used as possible hub genes, i.e., genes that connect to more than 10 other genes belonging to the genetic interaction network. Hub genes are critical since their dysregulation can not only potentially disrupt the pathway itself but also isolate other nodes.

### 3.5. Identification of Hub Genes

To identify the hub genes, we used two methods that were combined later. First, we identified the 45 most interacting DEGs at least present in two pathways, as shown in Figure 5. Out of the 16 pathways, 15 were included in this interaction to find any central pathway (discussed later). The prolactin pathway was filtered out due to the absence of any connecting gene to other pathways.

As can be seen, the resulting network contained 60 nodes and 115 edges (Figure 6a and Appendix A). In the second approach, we selected the previously identified 122 DEGs in Section 3.2 that were present in at least 5 or more pathways at the start of the analysis, as shown in Figure 3. The STRING database was used to construct the protein–protein interaction (PPI) network using these DEGs. This network revealed 95 nodes and 241 edges (Appendix A). The constructed network was then exported to Cytoscape, and the CytoHubba application was used to find any central DEGs. CytoHubba identified 13 DEGs, of which 2 (Cxcl10 and Ccl2) were up-regulated, and 11(Acta2, Cd34, Col1a1, Col1a2, Cxcl12, Eln, Igf1, Igf2, Itgam, Serpine1, and Thbs2) were down-regulated (Figure 6b). This PPI network contained 13 nodes and 34 edges. Co-expression of the key DEGs identified by CytoHubba was determined using STRING, which showed a strong interconnection among all the key hub genes (Figure 6c). Thus, all of the identified key genes by CytoHubba were also present in the identified significant pathways by GSEA.

To find hub genes, we combined the key DEGs resulting from both approaches (DEGs mentioned in Figure 6a,b). These 50 DEGs were submitted to STRING for any interconnections, and the generated PPI network consisted of 44 nodes and 142 edges. K-means clustering using STRING showed that this network could be divided into two main clusters (Figure 6d and Appendix A). Cluster A (in red) contained 20 DEGs (Calml3, Creb3l4, Dapk1, Dlg4, Ggt5, Ggt6, Grb10, Grin1, Gsta3, Gsta4, Igf2, Innpp5d, Mgs2, Myc, Pik3cg, Plcb1, Prkaa2, Rpsd6ka5, Wnt4, and Wnt5b). Cluster B (in green) contained 24 DEGs (Acta2, Alox5ap, Angpt2, Ccl2, Cd34, Chad, Col1a1, Col1a2, Col6a3, Col9a2, Cxcl12, Eln, Ibsp, Icam1, Ifit1, Igf1, Itga10, Itgam, Lama3, Pdgfd, Rasgrp3, Serpine1, Thbs2, and Vim). Both clusters were exported to Cytoscape after removing 6 DEGs (Gucy1b2, Il18r1, Nfatc2, Nfatc4, Nr4a3, and Pla2g3) not connecting to the main clusters, and the MCODE plugin was used to determine any significant module with highly interconnected regions. MCODE resulted in four significant modules (Appendix A), and the best module with the highest score was selected (Figure 6e). This module contained 12 DEGs that could be considered as key hub genes. Out of these, four (33.3%) were up-regulated (Angpt2, Ccl2, Icam1, and Myc), while eight (66.7%) were down-regulated (Acta2, Cd34, Col1a1, Col1a2, Cxcl12, Eln, Igf1, and Itgam), maintaining the overall down-regulatory effect on MMTV on the expression of key hub genes.

To further analyze the known function of the genes, these genes were submitted to the STRING database for clustering analysis via k-means, which resulted in three distinct clusters based on their functions (Figure 7a and Appendix A). The subsequent network comprised 12 nodes and 65 edges, with Cluster 1 comprising 5 genes (Angpt2, Ccl2, Cd34, Icam1, and Itgam). Functional enrichment analysis using GO showed that genes grouped in Cluster 1 are found in the extracellular region and are involved in cellular extravasation, leukocyte migration, stimuli response, cell differentiation, stress response, regulation of cell death, and immune response. Cluster 2 comprised four genes, including Acta2, Col1a1, Col1a2, and Eln, genes that belong mainly to the extracellular matrix and morphogenesis processes and are involved in protein digestion and absorption, focal adhesion, proteoglycans in cancer, inflammatory response pathway, and dysregulated miRNAs targeting insulin/ PI3-AKT signaling. Cluster 3 displayed three genes (Cxxl12, Igf1, and Myc), associated mainly with cell proliferation, transcription, breast cancer, transcriptional dysregulation in cancer, and proteoglycans in cancer. The CTDbase was used to find any involvement of these genes in animal diseases. As expected, all of these genes were involved in various diseases, such as cancer, diabetes, neoplasms, cardiovascular diseases, leukemia, and neurodegeneration (Figure 7b).

### 3.6. Identification of Central Pathway Dysregulated after MMTV Expression

Although by using GSEA, we found 6 pathways up-regulated and 25 down-regulated after MMTV expression, we selected the 16 top pathways on the basis of their strong statistics (Figure 4 and Table 3). Gene pathways interaction network analysis (Figure 5 and Figure 6a) showed the central role of PI3-Akt-mTOR pathway, and most of the genes were connected to this pathway. To further explore this pathway, we downloaded the already available pathway map of PI3-AKT-mTOR from KEGG comprising 359 genes and examined the overall gene expression, as observed in our study. We found 21 genes that were identified as DEGs in our study. These included: Angpt2, Chad, Chrm1, Col1a1, Col1a2, Col6a3, Col9a2, Creb3l4, Gm2436, Ibsp, Igf1, Igf2, Itga10, Lama3, Lpar4, Myc, Pdgfd, Pik3ap1, Pik3cg, Prkaa2, and Thbs2. Interestingly, most of these genes are key members of several pathways identified to be significantly dysregulated in our study. Except for Chrm1, Col6a3, Col9a2, and Gm2436, every gene overlapped five or more times in all pathways included in this study.

On the basis of DEGs identification and hub gene analysis, we modified the KEGG PI3-Akt-mTOR pathway map by showing the overall change in gene expression in this pathway (Figure 8). Our study shows that MMTV expression also significantly down-regulated genes involved in other pathways, such as insulin, ECM regulation, Wnt, and Ras signaling. The reduced expression of these genes. MMTV expression also induced the expression of Myc and Prkaa2 (*aka* Ampk) that may, in turn, induce cell cycle progression.

### 3.7. Prediction of mRNA-miRNA Interaction

MicroRNAs (miRNAs) are small non-coding RNAs involved in most biological processes by regulating mRNA expression [83]. Recent studies have well documented the regulatory role of miRNAs at both the transcriptome and/or proteome levels in various diseases [84]. Therefore, we predicted possible candidate miRNAs that may regulate expression of hub genes identified in this study. A total of 708 unique miRNAs were predicted for all hub genes using the miRDB database (see Appendix A). Further analysis showed that 58 miRNAs were predicted for at least three or more hub genes and mRNA-miRNA interaction map was created for these transcripts using Cytoscape. This map interaction contains 70 nodes and 192 edges (Figure 9).

Overall, most of the interactions were found for the Igf1 (n = 103), followed by Cxcl12 (n = 81), Col1a2 (n = 60), and Col1a1 (n = 59). The top predictions were found for mmu-miR-6951–3p and mmu-miR-7116–3p (n = 6), followed by mmu-miR-3102–3p, mmu-miR-466d-5p, mmu-miR-466k (n = 5), mmu-miR-126b-5p, mmu-miR-29a-3p, mmu-miR-29b-3p, mmu-miR-29c-3p, mmu-miR-466i-5p, mmu-miR-466l-5p, and mmu-miR-7119–3p (n = 4). These predicted miRNAs could be involved in the regulation of hub gene expression and may represent possible targets in cancer progression.

## 4. Discussion

MMTV is a well-known tumor-inducing pathogen in mice. In recent years, it has emerged as a potential pathogen in humans as well [35,36,37,38,39]. Therefore, it is important to study virus–host interactions to understand the host defense mechanisms at play and how the virus may evade such processes at the molecular level and spread within the host to cause disease. In the present study, we explored the host molecular pathways that may regulate the replication and potential pathogenicity of MMTV in normal mouse mammary epithelial cells, the key cells targeted by the virus for tumorigenesis. Our study is the first of its kind that has not only deeply analyzed changes in host gene expression upon MMTV expression but also identified key hub genes, downstream biological pathways, and gene regulators (miRNAs) involved in multiple cellular pathways affected by MMTV infection in the most relevant cells targeted by MMTV. Furthermore, our study presents predicted interactions of the miRNAs with identified hub genes, an aspect that points to the complex signatures induced upon MMTV infection. Thus, this study provides the foundational framework to understand the molecular circuitry involved in MMTV infection of mammary epithelial cells.

Our goal was achieved by conducting RNAseq that found 965 genes differentially regulated upon MMTV expression (Figure 2). Of these, 26% genes were up-regulated, while 74% were down-regulated upon MMTV expression, revealing a global down-regulation of gene expression induced by MMTV. To understand which molecular pathways were affected by the virus, gene ontology was performed, and DEGs were grouped in several biological pathways on the basis of the current literature. Most of the DEGs were involved in transport, regulation of cell proliferation, proteolysis, transcriptional regulation, cell differentiation, and signal transduction, which affected several molecular functions, including receptor binding, metal ion binding, catalysis, and hydrolase activities (Figure 3). Initially DEGs were grouped into 54 biological pathways on the basis of the available literature and database entries and then were restricted to 38, as we were interested in KEGG pathways for further analysis (Table 2; see Appendix A). Next, gene enrichment analysis was used to explore the role of these pathways upon MMTV expression. GSEA identified 31 pathways significantly enriched upon MMTV expression. These 31 pathways were further restricted to 16 on the basis of statistical values (Table 2). Of these, 3 pathways, including TNF, autophagy, and type II interferon were up-regulated, while 14 (including Wnt, Hedgehog, Focal adhesion, Rap1, Hippo, Egfr, Prolactin, PI3-Akt-mTOR, Ras, Metabolism, Inflammation, Estrogen, Glutathione, and Vegf) were observed to be down-regulated after MMTV expression (Figure 4 and Table 3). Gene–pathway interaction/networking map revealed that other than Prolactin, the remaining pathways were interconnected with each other via overlapping genes (Figure 5). Interestingly, most of the DEGs fell into multiple pathways, with 122 genes present in 5 or more pathways (Figure 3). Meanwhile, we also identified DEGs that were interconnecting significant pathways in this study (Figure 5 and Figure 6a). Of the 12 core hub genes, 4 were up-regulated (Angp2, Ccl2, Icam, and Myc), and 8 were down-regulated (Acta2, Cd34, Col1a1, Col1a2, Cxcl12, Eln, Igf1, and Itgam) (Figure 6e). The CTDbase was used to find any association of these genes with current diseases (Figure 7).

One may question why the HC11 cells were used for this analysis and not a more suitable in vivo mouse system. The HC11 cell line represents the simplest way to study the cellular/molecular changes after MMTV infection, especially since it retains the stem-cell-like property of primary mammary epithelial cells with the ability to differentiate into both ductal and luminal cells [46,47]. Normal mammary epithelial cells are also the final destination of MMTV infection in mice, where large amounts of virus are produced for passage to the progeny [3,12,13]. Consequently, these cells are the ones that eventually give rise to breast tumors as a result of insertional mutagenesis, where viral enhancer elements end up activating expression from growth promoting genes, such as Wnt1 and Fgf3 [25]. Under ideal circumstances, this study could have been carried out using a complex tissue environment, such as the infected mouse mammary gland, but considering these glands are composed of multiple cell types (epithelial (ductal, luminal), adipocytes, fibroblasts, lymphocytes (T-, B-, and NK-cells), and others) and undergo constant changes throughout the mouse lifespan (virgin, pregnant, lactating, and involution back to a non-lactating state [85]), studying the effect of virus infection in such a complex *milieu* with appropriate controls would have been difficult, to say the least, and would not have addressed what exactly happens in the mammary epithelial cells only. Hence, we chose the simplest cell line system to identify the response of the mammary epithelial cells to MMTV expression in the early stages of infection in a clear-cut manner without other confounding variables, which was the main aim of our study.

To further characterize our data, we compared our results with already published gene expression data in specific mouse models of breast cancer. Herschkowitz et al. [86] compared genetic profiles of 13 most commonly used mouse breast cancer models with human breast tumors and observed that not a single mouse model recapitulated all the expression features of a given human subtype. However, the shared features could provide a framework for improved understanding of MMTV-induced tumorigenesis. Therefore, we looked at the various mouse models available and compared our results with two more relevant models for which appropriate data were available.

In the first model, Cai et al. performed gene expression analysis on mammary tumors in the PyMT mouse model, in which mammary tumors were induced by expressing the polyoma middle T (PyMT) oncoprotein from the MMTV promoter [87]. The PyMT mouse model has been widely used to study breast cancer and shares similar transcriptional and morphological features with human breast cancer [86,88]. The authors analyzed mRNAseq data from four distinct stages of mammary tumor development, starting from hyperplasia, adenoma, early carcinoma, to late carcinoma. Characterization of the specific DEGs in the PyMT mouse model revealed that many of these genes were also differentially regulated in our study. Appendix A represents the up- and down-regulated genes in our study that were also observed in the PyMT mouse model. Moreover, six key hub genes identified by our study (Acta2, Cd34, Col1a1, Col1a2, Cxcl12, and Igf1) were also significantly dysregulated in the PyMT mouse model. Cai et al. found down-regulation of ECM, focal adhesion, insulin, PPAR, and metabolic pathways. We also observed similar down-regulation of these pathways upon MMTV infection, except ECM that was not included in our study (Table 2). Moreover, upregulation of p53 and cell cycle pathways during tumorigenesis documented by Cai et al. was also observed in our study in MMTV-infected HC11 cells (Table 2). Furthermore, categorization of common DEGs according to their association with KEGG pathways revealed similar down-regulation of many of these pathways in both our study and the Cai et al. study (Figure 10a). Considering that Cai et al. observed that the expression profile of most of these DEGs remained similar throughout cancer progression among the four stages of tumor progression, they suggest that alterations to gene expression important to carcinogenesis happen early on and are maintained throughout tumor progression.

Interestingly, similar to our results, they also observed an overall global down-regulation of most genes (63–69%) in the four stages, changes that actually started in the hyperplasia stage and continued during the later stages of tumor development [87]. Expression of one of the targets of their hub genes was the maintenance methyltransferase, DNMT1, that was significantly up-regulated consistently in the four stages of tumor progression analyzed. Not so surprising, we made a similar observation that MMTV up-regulates expression of the de novo methyltransferase, DNMT3L, an observation that may explain the global down-regulation of gene expression observed in our data (Figure 2 and unpublished observations).

This unexpected similarity in results between our cell line and their mouse model suggests that similar to the PyMT oncogene, MMTV expression may initiate changes in the HC11 cells that later predispose these cells to cell transformation. While MMTV does not encode an oncogene, earlier studies have implicated MMTV Gag and Env proteins with possible oncogenic properties, which may initiate these changes in the infected HC11 mammary epithelial cells early on [9,34,89]. This similarity could also be due to the stem-cell-like property of the HC11 cells used, as has been documented by Thrasyvoulou et al. 2020. They showed that retrotransposition of HC11 and not the non-tumorigenic C127 mammary epithelial cells by the VL30 retrotransposon led to acquisition of mesenchymal features by these epithelial cells, increased their cancer-stem-cell-like property, leading to tumorigenesis in BALB/c mice [90]. Together, these observations suggest that MMTV-induced molecular changes in HC11 cells could represent early changes that may be important for cell transformation and ultimate tumorigenesis. Thus, our findings reinforce the conclusion that HC11 cells represent a valuable in vitro model system to study gene regulation after MMTV infection.

In addition to the PyMT mouse model, the MMTV-Wnt1 mice are also used for studying breast cancer development [91]. This model is especially relevant to MMTV biology since up-regulation of Wnt1 expression via MMTV insertion is one of the main mechanisms of MMTV-induced tumorigenesis in mice [25]. Wnt gene family members are responsible for not only regulating cell growth and differentiation but also renewal of stem cells. Wnt signaling happens canonically through Frizzled (Fzd) and low-density lipoprotein (LDL) receptor-related proteins (LRP) and non-canonically through calcium, Jnk, and Src, resulting in developing two types of tumors, Wnt1 early, and Wnt1 late. Wnt1 early tumors show higher expression of genes from both the canonical and non-canonical Wnt signaling and EGFR signaling pathways compared with Wnt1 late tumors [92,93,94]. Comparison of our data with these studies revealed that our results were more similar to those observed in the Wnt1 early tumor stage than Wnt1 late tumors. A closer comparison of 76 genes from the canonical and non-canonical Wnt1 signaling pathways that were differentially regulated in the Wnt1 early tumors revealed that 50% of these genes were expressed in a similar manner to our study, while 41% were down-regulated, and only 9% were up-regulated, revealing that most of these genes were either similarly expressed or down-regulated in MMTV-expressing cells (Figure 10b and Appendix A represents expression profile of these genes in our study). Interestingly, there was a significant down-regulation of inhibitors of Wnt signaling, Nkd1/2, Sfrp1/2/5, and Dkk2/3/4, and up-regulation of Wnt activation genes, Frat1 and Frat2, in our study compared with Wnt1 early tumors, but expression of the Dishevelled family members (Dvl) that is activated by Wnt signaling was not changed (Figure 10b) [95]. This observation suggests that MMTV may induce Wnt signaling by a combination of down-regulating its inhibitors and up-regulating its activators. Furthermore, Nkd2 expression has been found to activate EGFR ligand. The EGFR pathway was found to be activated in Wnt1 early tumors [96], and as expected, down-regulated in our study. Meanwhile, expression of other EGFR family oncogenes, Erbb2 (HER2) and Erbb3 (HER3), was also down-regulated in our study (Appendix A). Overall, these results suggest that upon expression, MMTV promotes Wnt signaling irrespective of its well-known ability to activate it via insertional mutagenesis, perhaps as an early response to cell transformation.

To further investigate the precise expression of our hub genes, we then analyzed and compared their expressions in the RNAseq studies conducted on humans with breast cancer since; unfortunately, these data were not available for mouse mammary tumors. Therefore, the human Cancer Genome Atlas (TCGA: https://www.cancer.gov/about-nci/organization/ccg/research/structural-genomics/tcga, accessed on 16 April 2022) was searched for the expression of the hub genes identified in our study in human breast cancer patients. TCGA contains RNAseq data of 10,967 cancer patients, of which 1,084 are breast cancer patients having cancer at different stages (Appendix A). First, we examined the overall expression of the hub genes in all cancers and found that these genes were significantly expressed in most cancers, whether they were specific to the breast tissue or otherwise (Figure 10c). Then, the data were filtered for breast cancer patients with the ID “BRCA”. Our analysis revealed that most of the human breast cancer samples were of cancer stage M0 (no evidence of distance metastasis), with variable tumor stages, mostly T1 (tumor is <2 cm across) and T2 (tumor is >2 cm across). The expression profile of hub genes was variable in human samples; however, expression of Cd34, Icam1, Acta2, Eln, Cxcl12, Igf1, and Myc was consistent with and similar to our study (Figure 10d). Interestingly, the expression profile of Ccl2, Col1a1, Col1a2, and Itgam was very close to most of the breast cancer samples, with a degree of variability among samples. Although little is known about their role in terms of MMTV infection, we may map their expression profile changes during MMTV infection through observations made previously for other viruses. Most of the hub genes identified in this study have direct association with cellular immune response upon pathogen infection. We observed an up-regulation of type II interferon pathway and down-regulation of genes associated with inflammation and immune response. This agrees with the general observation that the first objective of any pathogen is to bypass or dysregulate cellular immune systems [97,98]. Appendix A describes a detailed relevance of the identified hub genes in this study to the viral infection and MMTV biology.

Although many biological pathways, including apoptosis, autophagy, cell cycle, Tnf, interferon, p53, Notch, ECM, Wnt, Hedgehog, focal adhesion, and Egfr, were dysregulated after MMTV expression, we recognize PI3-Akt-mTOR as one of the key pathways connecting many other pathways. To further explore the changes in PI3-AKT-mTOR pathway after MMTV expression, we analyzed this pathway from the KEGG database by superimposing our data on its map (Figure 8). This pathway contains most of the genes already defined as DEGs and hub genes in our study (Figure 5 and Figure 7). Activation of this pathway has been well known during viral entry and in cancers, especially breast cancer [99,100]. Activation of this pathway not only controls cell survival, metabolism, growth, and motility but also induces resistance against anti-cancer therapies by inducing tumor growth [101]. Inhibition of this pathway has been suggested as an important step not only to control viral entry but also cancer progression [100,102]. Various stimuli, including growth factors, cytokines, and hormones could activate the PI3K cascade. In most cancers, PI3K can be regulated by decreased expression of Pten (a direct antagonist of PI3K) [103] or induced expression of PIK3CA (a gene encoding PI3K). In our study, expression of Pten was above normal after MMTV infection, showing that down-regulation of PI3K pathway was from some other stimuli. We also found significant down-regulation of PIK3CG, the catalytic subunit of PI3K. Down-regulation of PIK3CG can impede the PI3K/AKT/mTOR pathway that has been suggested as a possible therapy in cancer [104]. We also found several genes as DEGs that may regulate PI3K signaling upon activation (Figure 7). Most of these DEGs were down-regulated in our study, including Igf1, Igf2, Pdgfd, Col1a1, Col1a2, Bcap, Pik3cg, Rasgrp3, Wnt4, and Fasl. PI3K activation blocks Gsk3b signaling that further blocks Myc expression. Thus, enhanced expression of Myc in our study could be a reason for reduced PI3K signaling.

While we found most of the important cellular pathways of the host down-regulated upon MMTV expression, some pathways were also up-regulated. These pathways are also important in cancer progression and were associated with cell cycle arrest, cell repair (e.g., p53 pathway) [105], apoptosis [106], tumor suppression, Notch pathway [107]), angiogenesis, host immune response (Tnf signaling) [108], type II interferon signaling [109]), cellular immunity, stress response, and inflammation (NFĸB signaling [110]). Thus, manipulation of these pathways by MMTV in the host could provide advantages to the virus to escape from the host innate and induced immune response and cause pathogenesis, aspects that need further exploring. Moreover, we also predicted miRNAs that may regulate the expression of important genes dysregulated during MMTV entry. Viruses are able to regulate host miRNA environment that, in turn, can alter host cellular mRNAs to achieve their targets [111]. In future studies, we hope to identify such miRNA dysregulated upon MMTV infection.

In conclusion, we find that upon MMTV expression in mammary epithelial cells, most of the DEGs and pathways dysregulated were associated with host defense mechanisms. Disruption of these pathways by MMTV could be taken as a possible mechanism to bypass the host immune system mounted by the cells to tackle MMTV infection. While a limitation of this study is that it was performed in a cell line, conducting this study in mammary epithelial cells isolated from MMTV-infected mammary glands would not have been possible due to technical issues of how to isolate these cells and which cell type to focus on since they differentiate and de-differentiate with time and hormonal status, showing a heterogenous phenotype at the molecular and cellular levels. Despite this limitation, we observed a similar down-regulation of gene expression pathways in HC11 cells after MMTV expression as those observed in the PyMT mouse model, and comparison with Wnt1 mouse model provided further insights into how MMTV expression may activate the Wnt1 signaling to predispose the mammary epithelial cells to cell transformation. Finally, most of the hub genes identified in our study were also found to play a role in human breast cancer. The identified hub genes post MMTV expression and predicted miRNAs targeting these hub genes may play critical roles in the progression of MMTV infection and cell transformation and could be used as possible diagnostic and disease progression biomarkers.

## Figures and Tables

**Figure 1 viruses-15-01110-f001:**
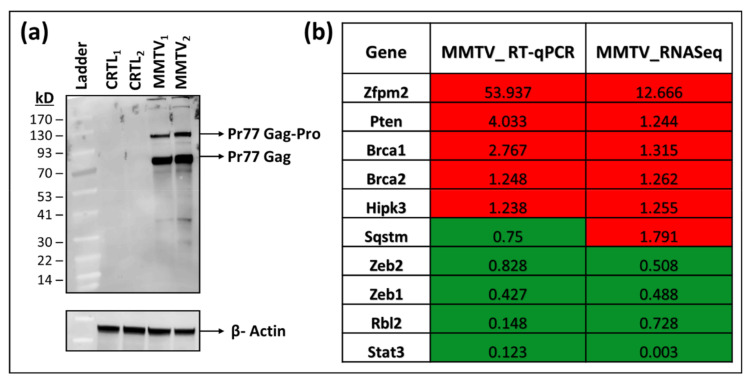
Establishment of HC11 cells stably expressing mouse mammary tumor virus (MMTV) and validation of their RNAseq data. (**a**) Western blot analysis of MMTV Gag gene expression in normal mouse mammary epithelial HC11 cells stably expressing MMTV. β-actin was used as the loading control. (**b**) RT-qPCR validation of RNAseq outcomes in MMTV expressing HC11 cells. The relative quantitative (RQ) values (MMTV/CTRL) obtained from RT-PCR data were compared with the relative (MMTV/CTRL) expression values of individual genes obtained from RNAseq data. The genes with value < 1 represent down-regulated genes, while those with value > 1 represent up-regulated genes, respectively, after MMTV expression. The red highlighted boxes depict up-regulated, while green boxes show down-regulated genes in cells expressing MMTV when compared with control HC11 cells.

**Figure 2 viruses-15-01110-f002:**
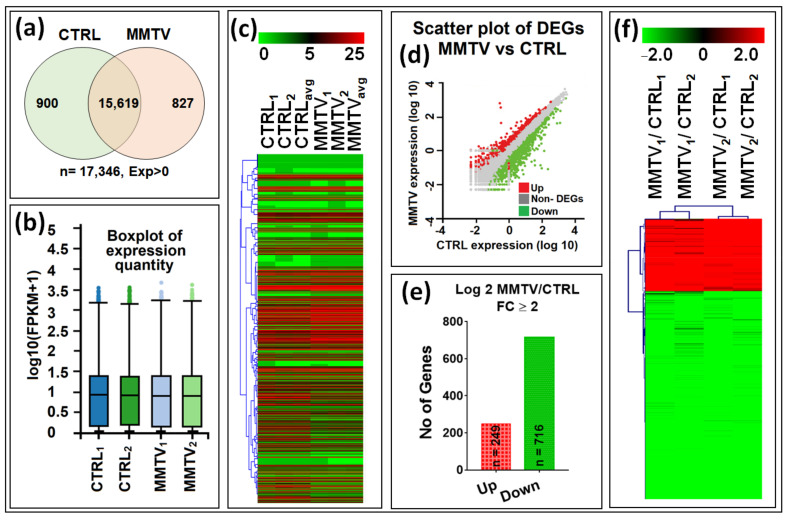
Summary of the gene expression analysis of the RAW and DEG data. After sequencing, a total number of 17,346 raw transcripts were retrieved. In this study, 2 biological replicates of each group were sequenced, and the average was taken to ensure the similarities within groups. For DEG analysis, cross-sectional comparison was performed for each group. Initially, fully automated Dr. Tom software from BGI was used to analyze the raw data as well as for the differential expression calculations. (**a**) Venn diagram showing the expression of total (n = 17,346), overlapping (n = 15,619), and unique transcripts (genes) between CTRL (n = 900; green circle) and MMTV (n = 827; pink circle) groups. (**b**) Tukey box plots showing the comparison among the sample level distribution of gene expression data post normalization for 4 samples. (**c**) Heatmap of the hierarchal clustered RAW data representing expression profile of 17,346 transcripts in all groups. (**d**) Scatter plot of DEGs showing the expression profile of DEGs and non-DEGs. Red dots show up-regulated genes, whereas green dots show repressed or down-regulated genes. The remaining (gray) were classified as non-DEGs. (**e**) Mapping of differentially expressed genes. Up- (red) and down- (green) regulated genes with in each comparison included in this study with fold change (FC) ≥ 2. (**f**) Heatmap showing the hierarchical clustering patterns of the 965 differentially expressed genes with FC ≥ 2 among all groups.

**Figure 3 viruses-15-01110-f003:**
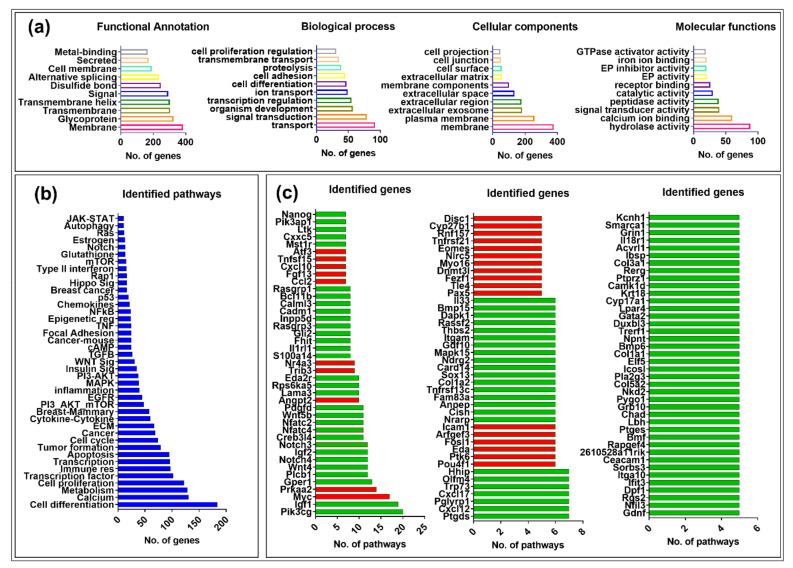
Gene ontology (GO) analysis of the differentially expressed genes (DEGs). (**a**) Classification of the DEGs based on their functional annotation, biological processes, cellular locations, and molecular functions. The top 10 sub-categories based on the number DEGs in each category are represented in each graph. (**b**) Biological pathway classification of DEGs. The DEGs were grouped in different pathways on the basis of the information gathered from different online pathways sources, including KEGG, Wikipathways, and/or Reactome. Each bar shows the number of genes associated with the subsequent pathway. (**c**) Overlapping genes in multiple pathways. The red bars display up-regulated, while the green bars show down-regulated genes.

**Figure 4 viruses-15-01110-f004:**
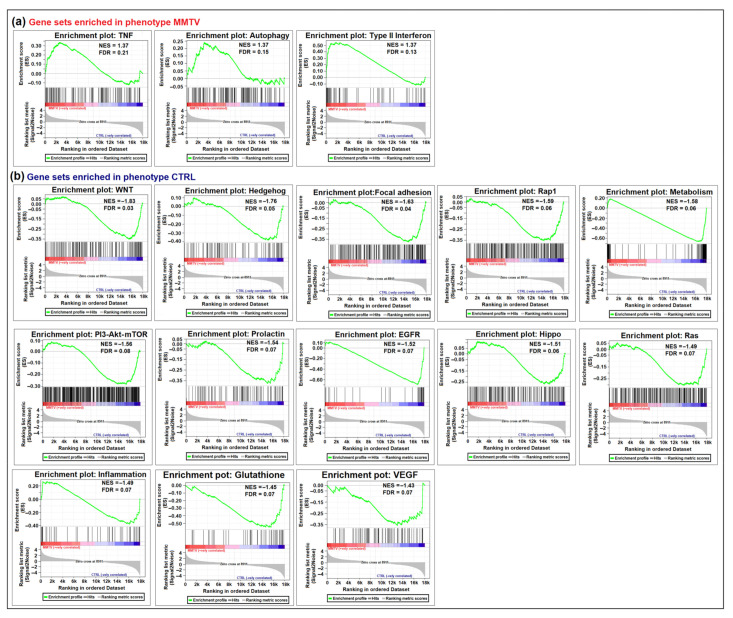
GSEA enrichment plots depicting negatively and positively enriched pathways after MMTV infection. A list of genes retrieved from KEGG pathways was used corresponding to the individual pathway listed in this study. (**a**) Gene sets positively enriched in phenotype MMTV. These included: TNF, Autophagy, and Type II interferon pathways. (**b**) Thirteen gene sets were negatively enriched in the MMTV phenotype or positively enriched in the CTRL phenotype. These gene sets included: Wnt, Hedgehog, Focal adhesion, Rap-1, Hippo, Egfr, Prolactin, PI3-Akt-mTOR, Ras, Metabolism, Inflammation, Glutathione, and Vegf signaling pathways. In each graph, the upper green slope corresponds to the enrichment score (ES), whereas the lower portion depicts the value of the ranking metrics moving down the list of ranked genes. The black vertical lines show the location of the genes in the gene sets in the provided list of genes.

**Figure 5 viruses-15-01110-f005:**
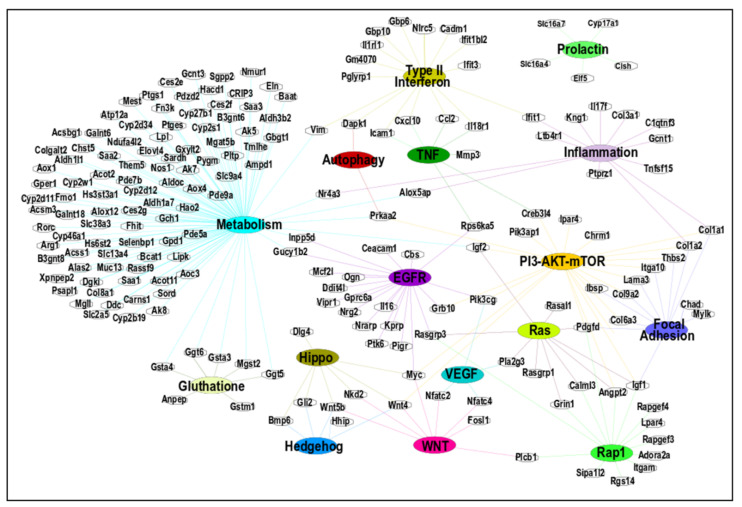
Networking among differentially expressed genes (DEGs) and pathways identified via GSEA. The top 16 pathways were identified using GSEA. Cytoscape was used to create the network using an organic layout. The network comprised 211 nodes and 258 edges. All genes included in this figure showed significant up- or down-regulated changes (FC > ±2).

**Figure 6 viruses-15-01110-f006:**
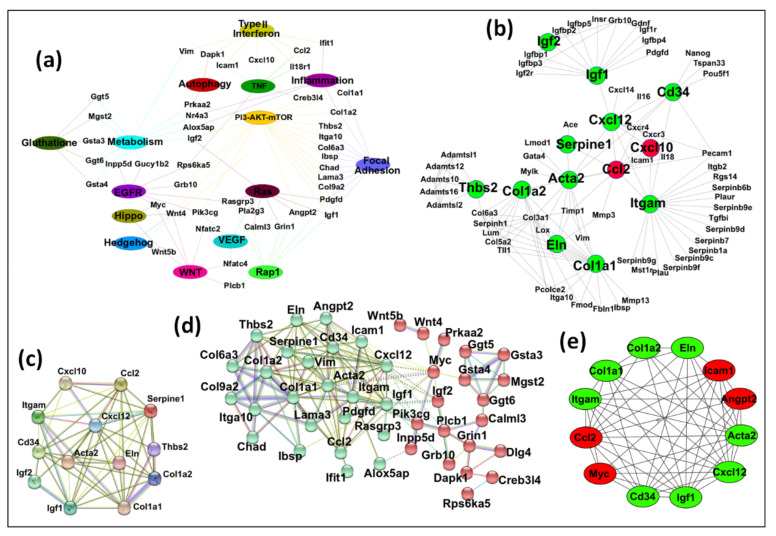
Identification of key hub genes. The gene–pathway and protein–protein interaction network analyses were conducted to find key hub genes, if any. (**a**) An interacting network among the significant pathways and the DEGs was created using Cytoscape. The network comprised 60 nodes and 115 edges. (**b**) The up- and down-regulated 13 hub genes identified by the Cytohubba plugin of Cytoscape through DEG PPI network identified by STRING. (**c**) The PPI network of 13 hub genes differentially expressed between CTRL and MMTV phenotype was constructed using STRING with default settings. (**d**) By combining the outcomes from (**a**,**b**) analyses, 50 DEGs were selected as potential candidates for hub genes. STRING divided them into 2 groups by means of k-means clustering (Cluster 1: red, Cluster 2: green). For STRING networks, the turquoise lines show known interactions from curated databases, the pink lines show experimentally determined interactions, and the yellow, black, and light blue lines show text-mining, co-expression, and protein homology interactions, respectively. The green, red, and blue lines show predicted interaction among gene neighborhoods, gene fusions, and gene co-occurrence, respectively. The dotted lines show interaction between two clusters. (**e**) The MCODE module of Cytoscape resulted in 12 most interconnected genes that were selected as key hub genes.

**Figure 7 viruses-15-01110-f007:**
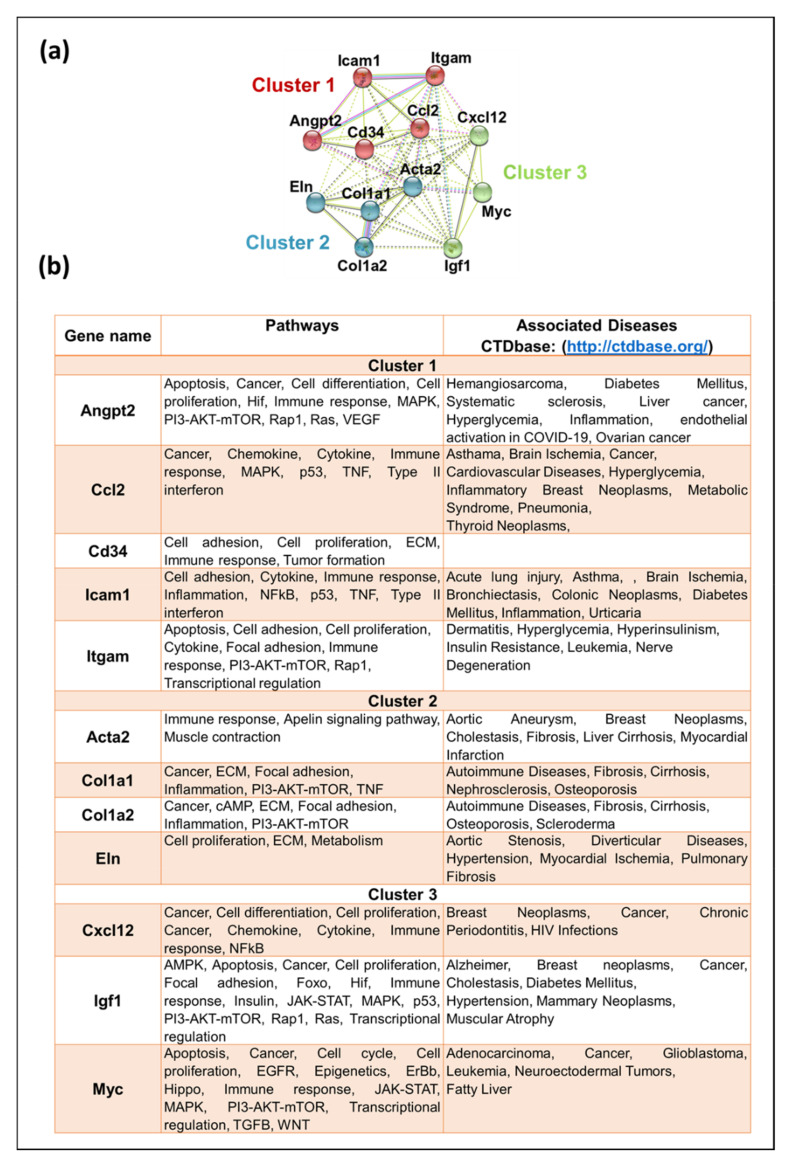
Functional analysis of key hub genes. (**a**) STRING was used to draw any functional similarities among key hub genes. K-means clustering using STRING resulted in 3 clusters, represented as red, blue, and green. The solid lines show interaction among the same group, while dotted lines show interaction among different clusters and sub-components. For STRING networks, the turquoise lines show known interactions from curated databases, the pink lines show experimentally determined interactions, and the yellow, black, and light blue lines show text-mining, co-expression, and protein homology interactions, respectively. The green, red, and blue lines show predicted interaction among gene neighborhoods, gene fusions, and gene co-occurrence, respectively. (**b**) Functional enrichment analysis and association of the key hub genes with known diseases. Genes were searched for the molecular pathways using KEGG, Wiki, and Reactome pathway databases as well as other gene databases. Association of these genes with diseases was searched in CTDbase, with only experiment evidence option.

**Figure 8 viruses-15-01110-f008:**
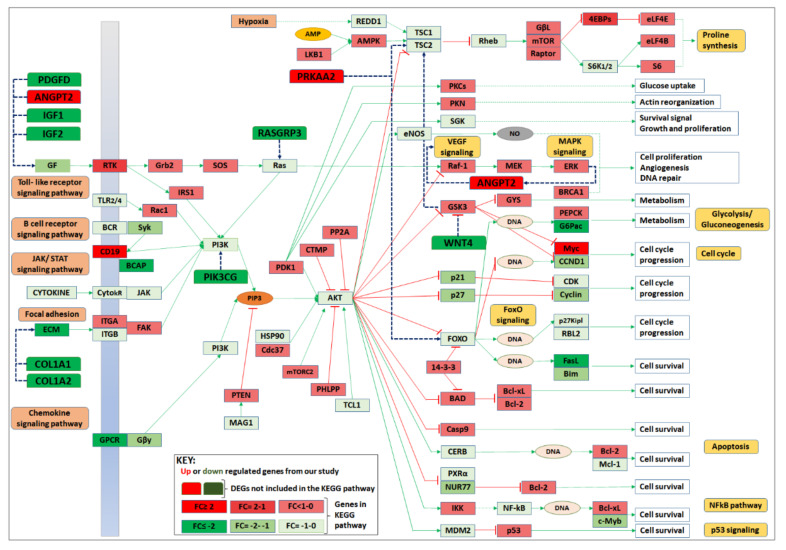
Expression analysis of genes involved in PI3-AKT-mTOR pathway post MMTV expression. The PI3-AKT-mTOR pathway was downloaded from the KEGG mouse pathways and was modified accordingly. The up-regulated genes are shown in red, while the down-regulated in green seemed to down-regulate Akt gene (a key gene of the PI3-AKT-mTOR pathway). Down-regulation of Akt could result in induced apoptosis, NFƙB, and p53 signaling that is also evident from our results, suggesting that MMTV may up-regulate these pathways, as suggested by GSEA analysis. This figure was created using KEGG "map04151 PI3K-Akt signaling pathway [68]" with copyright permission (230722).

**Figure 9 viruses-15-01110-f009:**
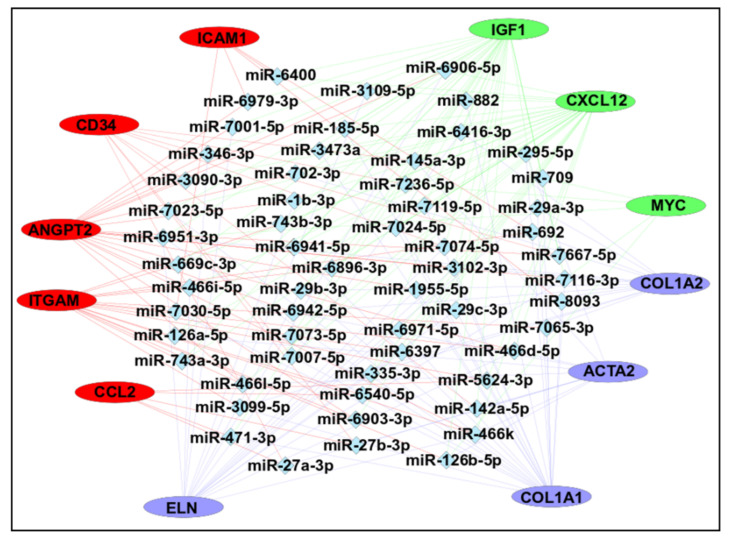
mRNA-miRNA interaction network. The mRNA–miRNA interaction map was constructed using Cytoscape with 12 hub genes and 58 predicted miRNAs. Every miRNA presented here showed interaction with at least 3 genes. The color of hub genes was selected based on the k-means clustering of the genes using STRING.

**Figure 10 viruses-15-01110-f010:**
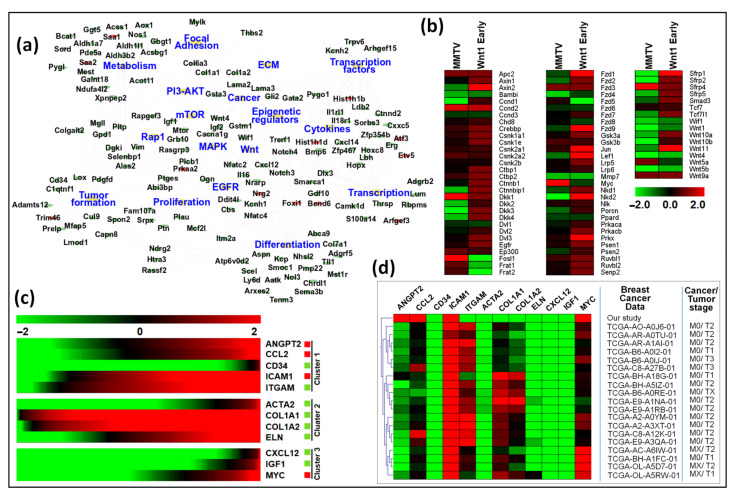
Gene expression comparison among our study, the PyMT, and Wnt1 tumor models, as well as human breast cancer data. (**a**) Networking among differentially expressed genes (DEGs) and pathways identified commonly in our study and the study by Cai et al. using PyMT mouse model [87]. The top 17 pathways were identified, sharing at least 5 or more common genes from both studies. Cytoscape was used to create the network using an organic layout. The network comprised 156 nodes and 271 edges. Red and green ellipse shapes represent up- and down-regulated genes with in KEGG pathways, respectively. The orange octagon shapes represent individual pathways. (**b**) Gene expression analysis between our study and Wnt1 early mouse model. Red and green boxes represent up- or down-regulated genes in both studies, respectively. (**c**,**d**) Functional analysis of key hub genes with panel (**c**) showing expression of key hub genes retrieved from RNAseq expression from profiles of all human breast cancer patients. The expression pattern of 12 hub genes in 10,967 human breast cancer samples was analyzed. (**d**) Clustering of hub genes and human breast cancer samples from the TCGA database. The expression pattern of the 12 hub genes in 1,084 samples was analyzed with respect to gene expression and tumor stage. The figure shows only the cluster containing our data. Red = up-regulated genes; Green = down-regulated genes.

**Table 1 viruses-15-01110-t001:** List primers and their details used in this study.

S. No.	Gene	Sequence	Forward (F)/ Reverse (R)	Stock Concentration	Working Concentration
1	Brca1	GGGGAAAAGGTAGGTCCAAAC	F	10 μM	100 nM
		CTGCTTCAGCATTTGACTCGT	R	10 μM	100 nM
2	Brca2	TCTTTCTCCGAGTATCAGGAAGT	F	10 μM	300 nM
		GCAGAAGTGTCAGTGAGAGTG	R	10 μM	300 nM
3	Hipk3	ATGGCCTCACAAGTCTTGGTC	F	10 μM	300 nM
		GCACTACCTTTCGTGGAAGGAT	R	10 μM	300 nM
4	Pten	TGGATTCGACTTAGACTTGACCT	F	10 μM	50 nM
		GCGGTGTCATAATGTCTCTCAG	R	10 μM	50 nM
5	Rbl2	CTGTGCTCCTTACACGACGG	F	10 μM	300 nM
		GCGGCTAACACGTATTCTTCA	R	10 μM	300 nM
6	Sqstm	AGGATGGGGACTTGGTTGC	F	10 μM	300 nM
		TCACAGATCACATTGGGGTGC	R	10 μM	300 nM
7	Stat3	CACCTTGGATTGAGAGTCAAGAC	F	10 μM	300 nM
		AGGAATCGGCTATATTGCTGGT	R	10 μM	300 nM
8	Zeb1	CGCCATGAGAAGAACGAGGAC	F	10 μM	300 nM
		CTGTGAATCCGTAAGTGCTCTTT	R	10 μM	300 nM
9	Zeb2	AAACGTGGTGAACTATGACAACG	F	10 μM	300 nM
		CTTGCAGAATCTCGCCACTG	R	10 μM	300 nM
10	Zfpm2	GAGCCCGAAAATCTGAGCTG	F	10 μM	50 nM
		GCCGACTCTGAATCTTCCTTTCT	R	10 μM	50 nM
11	β- Actin	TGTTACCAACTGGGACGACA	F	10 μM	300 nM
		CTGGGTCATCTTTTCACGGT	R	10 μM	300 nM

**Table 2 viruses-15-01110-t002:** Gene sets/pathways enrichment in phenotypes MMTV and CTRL. Rows shaded in gray show the significantly enriched pathways in both MMTV and CTRL phenotypes. ES: Enrichment Score, NES: Normalized Enrichment Score, NOM: Nominal, FDR: False Discovery Rate, FWER: Family-wise Error Rate.

Gene Sets	Size	ES	NES	NOM *p*-Value	FDR q-Value	FWER *p*-Value
Gene sets enriched in MMTV phenotype.						
TNF	110	0.34	1.37	0.222	0.215	0.170
Autophagy	137	0.24	1.37	0.000	0.159	0.170
Type II interferon	65	0.54	1.37	0.233	0.133	0.170
P53	70	0.23	1.25	0.000	0.169	0.451
Notch	57	0.24	1.23	0.240	0.164	0.570
Apoptosis	142	0.22	1.21	0.000	0.169	0.650
Cell cycle	121	0.23	1.13	0.245	0.284	0.950
NFƙB	97	0.31	1.08	0.422	0.325	0.950
ErBb	85	0.20	0.94	0.442	0.525	0.950
Gene sets enriched in CTRL phenotype.						
WNT	140	−0.35	−1.83	0.000	0.030	0.000
Hedgehog	79	−0.38	−1.76	0.000	0.053	0.070
Focal adhesion	195	−0.37	−1.63	0.000	0.045	0.100
Rap1	205	−0.35	−1.59	0.000	0.067	0.140
Metabolism	130	−0.66	−1.58	0.000	0.068	0.140
PI3−AKT−mTOR	395	−0.29	−1.56	0.000	0.080	0.140
Prolactin	76	−0.38	−1.54	0.000	0.076	0.140
EGFR	38	−0.69	−1.52	0.000	0.072	0.140
PI3−AKT	310	−0.30	−1.52	0.000	0.069	0.140
Hippo	148	−0.27	−1.51	0.000	0.073	0.140
Ras	221	−0.30	−1.49	0.000	0.070	0.140
Inflammation	46	−0.37	−1.49	0.000	0.071	0.140
Glutathione	46	−0.55	−1.45	0.000	0.075	0.180
VEGF	74	−0.36	−1.43	0.000	0.073	0.180
cAMP	190	−0.33	−1.39	0.000	0.086	0.350
Immune response	95	−0.47	−1.39	0.000	0.083	0.350
Estrogen	119	−0.34	−1.38	0.000	0.079	0.350
Calcium	219	−0.34	−1.32	0.000	0.097	0.450
AMPK	88	−0.22	−1.30	0.054	0.104	0.480
PPAR	70	−0.34	−1.27	0.000	0.107	0.480
mTOR	140	−0.19	−1.25	0.000	0.115	0.520
Chemokine	166	−0.24	−1.22	0.364	0.154	0.670
HIF−1	68	−0.19	−1.20	0.239	0.169	0.670
MAPK	300	−0.22	−1.20	0.000	0.164	0.670
FoXo	93	−0.23	−1.19	0.255	0.165	0.710
Insulin	130	−0.22	−1.07	0.352	0.339	0.930
TGFB	102	−0.23	−1.06	0.412	0.372	1.000
JAK−STAT	147	−0.26	−1.03	0.352	0.429	1.000
Cytokines	201	−0.25	−1.01	0.528	0.430	1.000

**Table 3 viruses-15-01110-t003:** List of enriched (core) genes (*p* value < 0.05) in each gene set corresponding to each group.

Sr. No	Pathway	Enriched (Core) Genes
MMTV Group		
1	TNF	Ccl2, Mmp3, Cxcl10, Icam1, Tnfrsf1b, Cxcl1, Fas, Tnfaip3, Ccl20, Vegfc, Creb5, Jag1, Lif, Traf5, Vcam1, Pgam5, Nod2, Ptgs2, Atf4, Il1b, Traf3, Tab1, Cxcl2, Csf1, Mlkl, NFƙBia, Pik3cb, Traf2, Ccl5, Cebpb, Rela
2	Autophagy	Prkaa2, Rragd, Rab39b, Smcr8, Uvrag, Bcl2, Ern1, Atg14, Sqstm1, Eif2s1, Irs1, Rptor, Atg2b, Ctsd, Dapk3, Igf1r, Deptor, Atg12, Pik3cb, Ppp2cb, Prkaa1, Irs2, Mapk10, Rragc, Nras, C9orf72, Akt1s1, Mapk8, Rraga
3	Type II interferon	Gvin1, Ccl2, Nlrc5, Gm4070, Pglyrp2, Ifit1bl2, Gbp6, Cxcl10, Gbp10, Ifit1, Ifi44, Ifit3b, Icam1, Gbp9, Ifit3, Isg15, Oas1a, Rsad2, Ifi211, Cgas, Cxcl9, Usp18, Gbp4, Gm4951, Stat2, Havcr2, Ifit2, Il1b
CTRL Group		
1	WNT	Porcn, Fzd2, Lef1, Camk2a, Nfatc1, Wif1, Wnt1, Wnt11, Tcf7l1, Sfrp5, Rac3, Plcb4, Wnt10a, Ctnnbip1, Wnt6, Fzd4, Camk2b, Vangl2, Sfrp1, Prickle2, Ccnd1, Plcb1, Nfatc4, Wnt4, Nfatc2, Wnt10b, Nkd2, Sfrp2, Wnt5b
2	Hedgehog	Evc, Arrb1, Bmp4, Stk36, Wnt1, Wnt11, Cdon, Hhip, Wnt10a, Gli1, Wnt6, Boc, Hhatl, Hhat, Gas1, Ccnd1, Gli2, Wnt4, Bmp6, Wnt10b, Wnt5b
3	Focal adhesion	Flnc, Pip5k1c, Col6a2, Itgb1, Chad, Col6a1, Actg1, Erbb2, Pak1, Thbs2, Lamc3, Itga1, Itgb6, Mylk2, Vegfd, ln2, Pik3r3, Myl9, Mylpf, Lamc1, Lamc2, Col9a3, Mylk4, Pik3cd, Itgb3, Itga6, Col4a5, Rac3, Col9a2, Tnn, Col9a1, Cav2, Itga2, Pdgfra, Ibsp, Pdgfc, Pdgfb, Vtn, Cav1, Tnc, Pdgfrb, Ccnd1, Col4a4, Itga10, Lama3, Col1a2, Lama2, Col1a1, Col6a3, Igf1, Pdgfd, Mylk
4	Rap1	Efna3, Pard6g, Vegfa, Pfn2, Plcb3, Magi1, Tek, Adcy1, Plcg1, Itgb1, Efna4, Actg1, Mapk12, Dock4, Vegfd, Tln2, Lpar1, Fgfr2, Pik3r3, Itgb2, Afdn, Ralgds, P2ry1, Fgfr3, Adcy8, Lat, Lpar5, Itgal Cnr1, Pik3cd, Map2k6, Itgb3, Tiam1, Angpt1, Rac3, Plcb4, Adora2b, Rgs14, Fyb, Pdgfra, Pdgfc, Adcy5, Pdgfb, Kitl, Pdgfrb, Lpar4, Rapgef4, Plcb1, Sipa1l2, Calm4, Rapgef3, Rasgrp3, Igf1, Calml3, Itgam, Pdgfd, Grin1
5	Metabolism	Sult4a1, Btn1a1, Adam8, Selenbp1, Cyp46a1, Pygl, Slc13a4, Dcxr, Bcat1, Vim, Ggt1,Gper1, Igf2, Dpep1, Ndufa4l2, Cbr2, Pde5a, Ggt5, Gsta4,Nos1, Pdzd2, Slc38a3, Aox1, Gpd1, Galnt18, Psapl1, Sardh, Mgat5b, Ak5, Gucy1b2, Aldh1a7, Gbgt1, Gcnt3, B3gnt8, Cyp2b19, Cyp2s1, Ptgs1, Acss1, Pltp, Inpp5d, Sgpp2, Carns1, Gsta3, Fhit, Ces2g, Alas2, Ptges, Sord, Fn3k, Mgst2, Dgki, Mest, Ces2f, Pde7b, Acsbg1, Aldh1l1, Col8a1, Atp12a, Pde9a, Cyp2d11, Hao2, Chst5, Ggt6, Colgalt2, Aldoc, Rorc, Ces2e, Them5, Cyp2d34, Lipk, Arg1, Mgll, Galnt6, Alox12, Xpnpep2, Tmlhe, Alox5ap, Aldh3b2, Gxylt2, Rassf9, Acsm3, Acot11, Cyp2w1, Ddc, Slc9a4
6	PI3-AKT-mTOR	Lamc3, Itga1, Ulk3, Itgb6, Ntf5, Myb, Vegfd, Hsp90b1, Prr5, Fgfr2, Pik3r3, Fzd2, Rxra, Cab39l, Erbb3, Creb3l2, Fgfr3, Ppp2r3a, Wnt1, Lamc1, Nr4a1, Wnt11, Lamc2, Ghr, Pkn3, Col9a3, Pik3cd, Gnb4, Gng11, Itgb3, Itga6, Bcl2l11, Col4a5, Angpt1, Cdkn1a, Col9a2, Tnn, Sgk2, Wnt10a, Col9a1, Itga2, Pdgfra, Ibsp, Wnt6, Gng7, Pdgfc, Pdgfb, Fzd4, Eif4e1b, Vtn, Prlr, Gngt2, Igf2, Syk, Tnc, Ppp2r2c, Pdgfrb, Ccnd1, Col4a4, Itga10, Chrm1, Lama3, Col1a2, Lama2, Creb3l3, Creb3l4, Wnt4, Pik3cg, Grb10, Col1a1, Pik3ap1, Wnt10b, Col6a3, Igf1, Pdgfd
7	Prolactin	Socs1, Mapk12, Thrsp, Pik3r3, Socs3, Socs2, Pik3cd, Btn1a1, Slc30a2, Pip, Prlr, Ccnd1, Cish, Cyp17a1, Elf5, Slc16a7, Slc16a4
8	EGFR	Cav2, Reps2, Pdgfc, Prlr, Cav1, Ddit4l, Nrarp, Rps6ka5, Ceacam1, Gucy1b2, Il16, Inpp5d, Pik3cg, Grb10, Rasgrp3, Kprp, Mcf2l, Pigr, Ogn, Gprc6a, Cbs, Vipr1
9	Hippo	Birc2, Prkcz, Crb2, Fzd1, Pard6g, Wnt7b, Llgl2, Serpine1, Actg1, Tgfb1, Smad3, Tgfbr2, Gdf5, Bmp4, Fzd2, Itgb2 Lef1, Tgfb2, Wnt1, Tead3, Wnt11, Tcf7l1, Dlg2, Ctnna3, Wnt10a, Wnt6, Fzd4, Ppp2r2c, Ccnd1, Gli2, Wnt4, Bmp6, Wnt10b, Nkd2, Dlg4, Wnt5b
10	Ras	Plcg2, Calml4, Efna5, Pik3r2, Gngt1, Fasl, Ralbp1, Fgf21, Rasgrp4, Ntrk1, Pla2g2f, Efna3, Vegfa, Foxo4, Tek, Gng2, Plcg1, Pld1, Efna4, Rasal2, Gm5741, Pak1, Ntf5, Vegfd, Fgfr2, Pik3r3, Afdn, Ralgds, Fgfr3, Lat, Pik3cd, Gnb4, Rgl2, Gng11, Tiam1, Pla1a, Angpt1, Rac3, Gab2, Pdgfra, Rasa4, Gng7, Pdgfc, Pdgfb, Pla2g2c, Gngt2, Igf2, Kitl, Pdgfrb, Calm4, Pla2g3, Rasal1, Pla2g4e, Rasgrp3, Igf1, Calml3, Pdgfd, Grin1, Rasgrp1
11	Inflammation	Il5, Lamc1, Lamc2, Fam13a, Ltb4r1, Vtn, Gcnt1, Col1a2, Col1a1, C1qtnf3, Ptprz1, Alox5ap, Col3a1
12	Glutathione	Gstt2, Gsta1, Gstm4, Gstm5, Gstk1, Gsto2, Mgst1, Idh2, Gstt1, Mgst3, Idh1, Gsta2, Gstm2, Ggt1, Ggt5, Gsta4, Gstm1, Gsta3, Anpep, Mgst2, Ggt6
13	VEGF	Mapk13, Ppp3cc, Akt3, Kdr, Pla2g6, Plcg2, Nos3, Pik3r2, Pla2g2f, Vegfa, Plcg1, Mapkapk3, Mapk12, Sphk1, Pik3r3, Nfatc1, Pik3cd, Rac3, Pla2g2c, Nfatc4, Pla2g3, Nfatc2, Pik3cg, Pla2g4e

## Data Availability

The raw and analyzed data (BioProject accession number: PRJNA915407) can be downloaded from the server for data re-analysis and further processing.

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
