# Peer review of "Global Down-regulation of Gene Expression Induced by Mouse Mammary Tumor Virus (MMTV) in Normal Mammary Epithelial Cells"

_viruses, 2023, doi:10.3390/v15051110_

Round 1

Reviewer 1 Report

Ahmad et al aimed to identify the genes and molecular pathways dysregulated by Mouse Mammary Tumor Virus (MMTV) in mammary epithelial cells. As a model, Ahmad et al used an immortalized murine mammary epithelial cell line, HC11 to compare transcriptional differences between the line itself and the line transfected with the plasmid containing hybrid MMTV (HYB MTV). The authors use a variety of computational techniques to outline not only the differentially expressed genes, but also the pathways and biological processes up- or down-regulated in the presence of HYB MTV. To expand upon their analyses, the authors also compared their gene expression data to that obtained from two murine tumor models and human breast cancer data. 

Major comments

            These studies are modeling MMTV infection using HC11 cells transfected with HYB MTV containing plasmid. It is unknown whether infection with MMTV would lead to similar changes in the transcriptomics of the HC11 cells. Furthermore, the authors should compare gene expression analysis of murine mammary gland tissue uninfected and infected with MMTV to get phytologically significant data.

·       Figure 2c: Clustering appears to have not be properly conducted. Clustering not done based on comparing expression in CTRL and in MMTV samples.

Minor comments

·       Figure 1b: There is a lack of a description of what the data depicts and how it was obtained (is it the average fold-change?).

·       Figure 2a: Unclear what “Exp” is.

·       Figure 3a: In “Functional Annotation” and “Cellular Components” plots, bar plots are cut off on X axes. If space is a concern, the X axes of “Biological Process” and “Molecular Functions” can be shortened to 100 genes.

·       Figure 9: Very blurry.

·       Table 1: Unclear what “S” and “AS” are in the Forward (F)/Reverse (R) column. Seems like entries should be “F” or “R”.

·       Table 2: Legend does not explain what contractions in table, such as “ES” and “NES”, are.

Author Response

Reviewer 1

Comments and Suggestions for Authors

Ahmad et al aimed to identify the genes and molecular pathways dysregulated by Mouse Mammary Tumor Virus (MMTV) in mammary epithelial cells. As a model, Ahmad et al used an immortalized murine mammary epithelial cell line, HC11 to compare transcriptional differences between the line itself and the line transfected with the plasmid containing hybrid MMTV (HYB MTV). The authors use a variety of computational techniques to outline not only the differentially expressed genes, but also the pathways and biological processes up- or down-regulated in the presence of HYB MTV. To expand upon their analyses, the authors also compared their gene expression data to that obtained from two murine tumor models and human breast cancer data. 

Major comments

            These studies are modeling MMTV infection using HC11 cells transfected with HYB MTV containing plasmid. It is unknown whether infection with MMTV would lead to similar changes in the transcriptomics of the HC11 cells. Furthermore, the authors should compare gene expression analysis of murine mammary gland tissue uninfected and infected with MMTV to get phytologically significant data.

Response: Thank you for the suggestion of comparing gene expression analysis of murine mammary gland tissue uninfected and infected with MMTV to get physiologically significant data. This is a good experiment that we actually are planning to conduct next.  However, for the present study, we specifically chose the normal mouse mammary epithelial cell line (HC11) model for this analysis since the “mammary epithelial cells” are the final destination of MMTV infection in the mice, which eventually give rise to the virus-induced breast tumors.  Using infected mouse mammary glands would not have answered our main aim of studying mammary epithelial cells only since these glands are composed of multiple cell types (epithelial [ductal, luminal], adipose, fibroblasts, lymphocytes (T- and B-cells, NK cells), and others and undergo constant changes throughout the mouse lifespan (virgin, pregnant, lactating, and back to non-lactating).  This would have defeated the purpose of addressing MMTV-induced changes in the “epithelial cell” component of the mammary gland.    Therefore, we chose the next best option of using the simplest HC11 cell line model to address only one variable at a time, the MMTV expression itself, to identify its effect on mammary cell transcriptome.  We have mentioned this aspect as a limitation of our study on lines 843-849 of the revised manuscript.

Unlike many other cell lines, HC11 cells are special since they have stem cells-like property which allows them to undergo differentiation upon prolactin induction.  In fact, it is this property that has allowed us to study, in parallel, the effect of MMTV infection in these cells upon differentiation in a parallel study (manuscript in preparation).  Thus, we believe that actually the HC11 system is physiologically a highly relevant and suitable system to study complex MMTV-host interactions step-by-step in a very systematic and deliberate manner with appropriate controls, making our study meaningful and appropriate. That is why the PubMed shows >480 studies published that have used this cell line to study mammary epithelial cell proliferation, differentiation, transformation, as well different aspects of MMTV biology.  

To address the second important point raised by the respected reviewer, while it would have been ideal to “infect” HC11 cells rather than create a stable cell line expressing the virus; however, we found that infection of these cells was inefficient (Kincaid et al., 2018; Virology 513:118-187).  This would have resulted in analyzing differential gene expression in a mixture of cells that were infected and uninfected, reducing our chances of picking up subtle changes in the expression of many genes.  Therefore, we decided to use stably-transfected HC11 cells where all our cells were expressing the virus.  Furthermore, since MMTV does not show superinfection resistance (Dzurius et al., 1999; Virology 263:418-426), these cells should still be undergoing natural multiple rounds of “infection” with the virus produced from the stable cells, thus creating an environment similar to the “infected” cell. This clarification has been incorporated on lines 276-283 of the revised manuscript.

  • Figure 2c: Clustering appears to have not be properly conducted. Clustering not done based on comparing expression in CTRL and in MMTV samples.

Response: This clustering represents raw gene expression from both control (CTRL) and MMTV groups and is not based on comparing expression in CTRL and MMTV samples. It was performed to know how the overall raw expression of individual genes (a total of 17,346 transcripts) resembled or was consistent within samples.  Figure 2f is where we show the clustering of only the 965 differentially-expressed genes (FC >2) based on whether they were up- or down-regulated.

Minor comments

  • Figure 1b: There is a lack of a description of what the data depicts and how it was obtained (is it the average fold-change?).

Response: In figure 1b, the relative quantitative (RQ) values (MMTV/CTRL) obtained from RT-PCR data were compared with the relative expression values obtained from RNAseq data (MMTV/CTRL). These values do not represent fold-change as fold-change is a logarithmic value. The value <1 depicts down-regulation, while >1 depicts up-regulation after MMTV expression. This has been clarified in the legend of Figure 1b of the revised manuscript.

 Figure 2a: Unclear what “Exp” is.

Response: “Exp” represents individual genes (n = 17,346) that were actually “expressed” as transcripts in our data with an expression value of > zero transcripts.  The green shaded circle represents the number of genes expressed in the control (CTRL) group only and the pink shaded circle represents the number of genes specifically expressed in the MMTV group only, while the no of genes in the overlapping region of the Venn diagram (a vast majority of the genes) were expressed in both groups.  This has been further clarified in the legend of Figure 2a of the revised manuscript.

 Figure 3a: In “Functional Annotation” and “Cellular Components” plots, bar plots are cut off on X axes. If space is a concern, the X axes of “Biological Process” and “Molecular Functions” can be shortened to 100 genes.

 Response: The x-axis of Figure 3a has been corrected in the revised manuscript.

 Figure 9: Very blurry.

 Response: Figure 9 has been replaced with a clearer image in the revised manuscript.

 Table 1: Unclear what “S” and “AS” are in the Forward (F)/Reverse (R) column. Seems like entries should be “F” or “R”.

 Response: Thanks for pointing this oversight on our part.  Both “S” and “AS” are abbreviations for “sense” and “anti-sense”, primers respectively. Now, these have been replaced in Table 1 by forward (F) and (R), as mentioned in the table header.

 Table 2: Legend does not explain what contractions in table, such as “ES” and “NES”, are.

 Response: The contractions used in the table have been explained in the revised manuscript.

Reviewer 2 Report

Limitations:

-       Overall, this study is descriptive and does not provide precise mechanism (s) of MMTV infection. There is an overwhelming number of genes/pathways that need to be integrated into a more cohesive mechanism.

-       Lack of experimental validations (serum/plasma cytokines) or genes will help validate the outlined results.

-       Discuss the role of T cell and B cell in MMTV infection

-       Are there any signatures associated with T /B cell activation/ function ?

Figures:

1-    Fig. 1b. RNA-Seq data:  Are these log fold changes or absolute values?

      Please specify in the figure legend.

2-    Fig. 2f: It does not see these genes are differentially expressed between controls and MMTV?

What is shown on the heatmap? Are these fold changes or absolute values

3-    Fig. 3b: what statistical test was used to evaluate the enrichment of pathways? Are these genes up or down? How about the pathways?

4-    Pathways in Table 2 are not all significant, Why?

5-    In Table 3, are all these leading genes significant? What is the p-value or the adjusted p associated with these genes?

6-    Are all the genes in Fig. 5 significant? Are these Up or down ? specify in the legend stats , p value, adjusted p , GSEA FDR , etc…?

7-    Row 618-622: Initially DEGs were grouped into 54 biological pathways based on available literature and database entries and then  were restricted to 38 as we were interested in KEGG pathways for further analysis (Table  2; see Supplementary Data). Next, gene enrichment analysis was used to explore the role of these pathways upon MMTV expression. GSEA identified 31 pathways significantly 622 enriched upon MMTV expression.

-       What is the overlap between the 54 and 31 pathways?

-       It is not clear why authors conducted two different enrichment analysis?

8-    Figure 10 panel a: Gene names are not clear , maybe have them in bold or a different color

9-    Figure 10 panel c: color gradient is counterintuitive, red is negative and green positive while in the remaining paper, it is the opposite. Make the color consistent

Author Response

Reviewer 2

 Comments and Suggestions for Authors

Limitations:

-       Overall, this study is descriptive and does not provide precise mechanism (s) of MMTV infection. There is an overwhelming number of genes/pathways that need to be integrated into a more cohesive mechanism.

 Response: We respect the views of the respected reviewer. This is the first ever attempt to analyze any changes at transcript levels after MMTV infection in mouse mammary epithelial cells using RNAseq. RNAseq results in much bigger data sets than usual lab experiments that are based on selected genes or pathways. Although we tried our best to cut short the data as much as possible without losing significant information, it is not possible to focus on a selected pathway(s), as happens in routine studies. This study is a comprehensive/complete analysis and should help other researcher focusing on specific aspects of MMTV-host interactions.

 While the respected reviewer is correct in the assertion that there are “an overwhelming number of genes/pathways” which is a direct consequence of the RNAseq analysis, we feel that we have provided additional value to our analysis by comparing the data to what has been observed in two different mouse models of breast cancer as well as what is observed in the human breast cancer patients.  Our results suggest interesting parallels in our system and what has been observed in these models, aspects which should be valuable to anyone studying MMTV biology.  For instance, we observed a global downregulation of genes via most likely activation of a DNA methyltransferase, a mechanism that has been observed in the PyMT model of breast cancer induction.  Similarly, the main mechanism ascribed to MMTV to induce mammary tumorigenesis is via insertional activation of growth promoting protooncogenes like Wnt1 in mammary epithelial cells. Comparison of our results with the Wnt1 mouse model revealed how MMTV expression could lead to activation of the Wnt1 pathway independent of insertional mutagenesis via activation of the inhibitors of the Wnt1 pathway and inhibition of its repressors.  Thus, our study reveals interesting new mechanisms that should be pursued experimentally for verification to find further insights into MMTV-induced tumorigenesis and breast cancer in general.

-       Lack of experimental validations (serum/plasma cytokines) or genes will help validate the outlined results

Response: This study was conducted using a mouse mammary epithelial cell line and not in mice, so it does not require any experimental validation using serum/plasma. However, we did compare our results with the available transcriptomics data from two important mouse models used for breast cancer studies and with human breast cancer data as well.

-       Discuss the role of T cell and B cell in MMTV infection

Response: This study was conducted in a mouse mammary epithelial cell line and we are not aware of any transcriptome study that has been done in T or B cells that have been infected with MMTV.  While we have already mentioned the role of T and B cells in MMTV life cycle in the introduction, discussing it further will only divert the focus of the paper from the mammary epithelial since we did not test either infected T- or B-cells.

-       Are there any signatures associated with T /B cell activation/ function?

Response: While we actually did not study MMTV-infected T or B cells in this study, we did find some differentially-regulated genes associated with these cells. These include, Inpp5d, Rasgrp3, Nfatc4, Pik3ap1, Nfatc2 and Pik3cg for B cell- and Nfatc4, Nfatc2, Rasgrp1 and Pik3cg for T cell-associated pathways. These genes are included in the Supplementary Material Sheet S7 and also added in the last paragraph of the Supplementary File S15.

Figures:

1-    Fig. 1b. RNA-Seq data:  Are these log fold changes or absolute values?

      Please specify in the figure legend.

Response: In figure 1b, the relative quantitative (RQ) values (MMTV/CTRL) obtained from RT-PCR data were compared with the relative expression values obtained from RNAseq data (MMTV/CTRL). These values do not represent fold-change as fold-change is a logarithmic value. The value <1 depicts down-regulation, while >1 depicts up-regulation after MMTV expression. This has been clarified in the legend of Figure 1 of the revised manuscript.

2-    Fig. 2f: It does not see these genes are differentially expressed between controls and MMTV? What is shown on the heatmap? Are these fold changes or absolute values

Response: Figure 2f is a heatmap of 965 differentially-regulated genes. It is not possible to name/ accommodate this large gene dataset in a single figure. Supplementary data S4 contains the name and expression data of all the genes included in this figure.  This heatmap contains gene expression in fold change (FC). This has also been clarified in the legend of Figure 2f in the revised manuscript.

 3-    Fig. 3b: what statistical test was used to evaluate the enrichment of pathways? Are these genes up or down? How about the pathways?

Response: Figure 3b represents classification of our 965 differentially-expressed genes (DEGs) shown in Fig. 2f into biological pathways using the tool DAVID at default settings.  They can be either up- or down-regulated.  DAVID uses both the Bonferroni and Benjamini statistical analyses to evaluate and subgroup the data to create enrichment scores. The biological classification was based on information gleaned from various online pathways sources, including KEGG, Wikipathways and/or Reactome.  Section 2.5 of the original manuscript describes the whole process in detail which is provided below as well for the convenience of the reviewer (lines 222-236):

2.5. Functional enrichment of gene ontology (GO) and pathway analysis

DAVID generated the list of genes involved in several biological pathways using KEGG (Kyoto Encyclopedia of Genes and Genomes: https://www.genome.jp/kegg/pathway.html) [67] and Reactome (https://reactome.org/) [68] pathway databases. We also searched the Wikipathways (https://www.wikipathways.org/index.php/WikiPathways) [69] for pathways associated with the DEGs. All of these databases have limitations and the DEGs whose GO or pathways were not defined using these databases were further searched using GeneCards (https://www.genecards.org/) [70] and Rat Genome Database for mouse specie (RGD: https://rgd.mcw.edu/wg/species/mouse/) [71]. The genes were sorted based on the combined data from all these sources and used for further analysis, unless otherwise stated. The genes without any verified information were shown as “uncharacterized”.

4-    Pathways in Table 2 are not all significant, Why?

Response: Table 2 was created using Gene Set Enrichment Analysis (GSEA) which gives both significant and non-significant pathways based on nominal p-values and FDR q-values. GSEA uses FDR q-value <0.25 for further analysis (https://software.broadinstitute.org/cancer/software/gsea/wiki/index.php/FAQ#Why_does_GSEA_use_a_false_discovery_rate_.28FDR.29_of_0.25_rather_than_the_more_classic_0.05.3F). Due to this reason, some pathways with p value>0.05 were also selected for further analysis.   This has also been clarified on lines 243-248 of the revised manuscript.  

5-    In Table 3, are all these leading genes significant? What is the p-value or the adjusted p associated with these genes? 

Response: The genes included in this table have p<0.05. This has been included in the header to the table of the revised manuscript.

6-    Are all the genes in Fig. 5 significant? Are these Up or down ? specify in the legend stats , p value, adjusted p , GSEA FDR , etc…

Response: Yes, all genes included in the Figure 5 are significant with FC>±2. These genes were derived from GSEA analysis and the statistical analysis of these pathways have been already discussed in Tables #2 and 3.  This has been clarified in the legend of Figure 5 in the revised manuscript.

7-    Row 618-622: Initially DEGs were grouped into 54 biological pathways based on available literature and database entries and then  were restricted to 38 as we were interested in KEGG pathways for further analysis (Table  2; see Supplementary Data). Next, gene enrichment analysis was used to explore the role of these pathways upon MMTV expression. GSEA identified 31 pathways significantly enriched upon MMTV expression.

-       What is the overlap between the 54 and 31 pathways?

Response: These 31 pathways were the part of initially-selected 54 pathways.

-       It is not clear why authors conducted two different enrichment analysis?

Response: In this manuscript, only one enrichment analysis was done on the 38 selected pathways (as described in Table 2) using data from KEGG database (see Section 2.6 of the original manuscript).   The reviewer may be confusing the shortlisting of the 54 biological pathways into 38 based on retrieval of data from KEGG database with the 31 pathways that resulted from the GSEA analysis shown in Table 2.

8-    Figure 10 panel a: Gene names are not clear , maybe have them in bold or a different color 

Response: Unfortunately, Figure 10 is a little bit compressed in the PDF version of the manuscript; however, we have improved its resolution in the revised version of the manuscript.

9-    Figure 10 panel c: color gradient is counterintuitive, red is negative and green positive while in the remaining paper, it is the opposite. Make the color consistent

Response: We are thankful to the respected reviewer for pointing out this discrepancy. Figure 10c has been re-set accordingly in the revised manuscript.

Reviewer 3 Report

This is an interesting paper and the authors are to be congratulated on a very in depth and detailed study. However, and unfortunately, a major caveat is that only one MMTV infected HC11 cell clone (and 1 non-infected clone) appear to have analysed. Indeed, it would have been better to have examined 2 or more clones rather than two independent passages of the MMTV infected and control cells (as the authors did) to rule out any effect of clonal variation. Simply put, how can the authors be sure that all of the gene expression changes they observed are associated with MMTV infection and not clonal variation? This raises questions about the validity of the result from this study.  The authors should consider at least a partial analysis for verification using another infected clone. This limitation needs to be addressed when discussing the data.

Other points to consider include:

1)    The authors used the Shackleford cloned provirus – this virus has been shown to productively infect mouse cells whereas the provirus used by Indik et al has been shown to productively infect both mouse and human cells (Indik et al., 2005 Mouse mammary tumor virus infects human cells. Cancer Research 65: 6651-6659) and that the infection is productive in that new particles are produced from the infected cells (Indik et al., 2007 Rapid spread of mouse mammary tumor virus in cultured human breast cells. Retrovirology 4: 73). The authors should mention that human cells have been shown to be productively infected by MMTV and cite these paper on line 97.

2)    In the manuscript, the authors refer to infected HC11 cells but in the methods section it sounds like transfected cell clones were generated. The authors should explain this. If transfected clones were used, then the findings might be different to those from infected cells.

3)    HC11 cells were chosen by the authors as a representative mouse mammary epithelial cell line but they should mention that HC11 cells a spontaneously immortilised cell line. There are other cell lines they could have chosen like EF43.

4)    The discussion on the comparison with transgenic mouse models should be weakened since these are somewhat artificial as compared to the real life situation. A particular concern is why the authors focus on the PyMT mouse model which uses mammary targeted expression of an oncogene that has not been shown to be involved in mammary tumorigenesis and why not other transgenic mouse models where oncogenes that have been implicated in breast cancer are targeted to the mammary gland such as MMTV-Neu and MMTV-H-Ras.

5)    The authors should discuss their data in light of the study by Thrasyvoulou et al., 2020 (VL30 retrotransposition is associated with induced EMT, CSC generation and tumorigenesis in HC11 mouse mammary stem‐like epithelial cells. Oncology Reports 44: 126–138).

6)    In general, the discussion is overly long and contains much speculation.

Minor Point

Lines 63 “Most of the endogenous strains of Mtvs are observed to be defective..”

The authors could reference Salmons and Günzburg (1987) Current Perspectives in the Biology of Mouse Mammary Tumour Virus. Virus Research 8: 81-102

Typo

We also observed induced NFkB pathway in our study in MMTV phenotype although it was not statistically significant It has been evident that viral infection dysregulates immune system and increased expression of Icam-1 and other adhesion molecules like integrins and angiogenic proteins

Author Response

Reviewer 3

Comments and Suggestions for Authors

This is an interesting paper and the authors are to be congratulated on a very in depth and detailed study. However, and unfortunately, a major caveat is that only one MMTV infected HC11 cell clone (and 1 non-infected clone) appear to have analysed. Indeed, it would have been better to have examined 2 or more clones rather than two independent passages of the MMTV infected and control cells (as the authors did) to rule out any effect of clonal variation. Simply put, how can the authors be sure that all of the gene expression changes they observed are associated with MMTV infection and not clonal variation? This raises questions about the validity of the result from this study.  The authors should consider at least a partial analysis for verification using another infected clone. This limitation needs to be addressed when discussing the data.

Response: We would like to thank the respected reviewer for appreciating our study.  The reviewer has raised a very important point that we would like to clarify.  Our MMTV stable cell line actually represents pooled colonies and not an individual clone.  Furthermore, as mentioned by the reviewer, we actually analyzed two independent passages (biological replicates) of the control and MMTV pooled stable cell lines in this RNAseq study (as shown in Fig. 1).  Thus, our gene expression data is not from just a clonal population.  To verify some of the effects of MMTV on gene expression that we were observing, we actually obtained fresh HC11 cells from ATCC and made an independent MMTV-expressing stable cell line.  Test of the expression of several genes of the prolactin pathway in this cell line confirmed the phenotype that we observed in our RNAseq analysis in this study.  In fact, we further went on to show a dose response on the expression of for example the beta casein gene with increasing levels of MMTV expression in different stable cell lines.  These data are part of a parallel study that we have conducted exploring the mechanism of the global repression of gene expression observed in this study that is being prepared for publication.  We have alluded to this study in our discussion of the PyMT model of tumorigenesis as well. Thus, we are quite confident that our results are valid and due to MMTV expression and not clonal variation.  This has been further clarified in the revised manuscript on line 313.

Other points to consider include:

1)    The authors used the Shackleford cloned provirus – this virus has been shown to productively infect mouse cells whereas the provirus used by Indik et al has been shown to productively infect both mouse and human cells (Indik et al., 2005 Mouse mammary tumor virus infects human cells. Cancer Research 65: 6651-6659) and that the infection is productive in that new particles are produced from the infected cells (Indik et al., 2007 Rapid spread of mouse mammary tumor virus in cultured human breast cells. Retrovirology 4: 73). The authors should mention that human cells have been shown to be productively infected by MMTV and cite this paper on line 97. 

Response: We thank the respected reviewer and the suggested text and references have been added to the revised manuscript (lines 97-98 of the revised manuscript).

2)    In the manuscript, the authors refer to infected HC11 cells but in the methods section it sounds like transfected cell clones were generated. The authors should explain this. If transfected clones were used, then the findings might be different to those from infected cells.

Response: This is a good point raised by the respected reviewer.  Ideally, we would have chosen to “infect” HC11 cells rather than create a stable cell line expressing the virus.  However, we found that infection of these cells was inefficient (Kincaid et al., 2018; Virology 513:118-187) and we would have ended up analyzing differential gene expression in a mixture of cells that were infected and uninfected, reducing our chances of picking up subtle changes in the expression of many genes.  Therefore, we decided to use stably-transfected HC11 cells where all our cells were expressing the virus.  Furthermore, since MMTV does not show superinfection resistance (Dzurius et al., 1999; Virology 263:418-426), these cells should still be undergoing natural multiple rounds of “infection” with the virus produced from the stable cells, thus creating an environment similar to the “infected” cell. That is why, we have used both terminologies interchangeably, but primarily the former one in the manuscript.  This clarification has been incorporated on lines 276-283 of the revised manuscript.

 3)    HC11 cells were chosen by the authors as a representative mouse mammary epithelial cell line but they should mention that HC11 cells a spontaneously immortalized cell line. There are other cell lines they could have chosen like EF43.

Response: As suggested by the respected reviewer, we have added the information that HC11 cells rose spontaneously as an immortalized cell line (lines 273-274 of the revised manuscript). 

While there are several other mammary epithelial cell lines available that could have been used for our study (such as EpH4 or EF43, as mentioned by the reviewer), we chose HC11 cell line due to its stem cell-like properties that make them differentiatable and the fact that not only is it prolactin responsive, but arose from the BALB/c mouse.  We (along with our collaborators) have used the BALB/c mouse model of mammary tumorigenesis along with the HYBMTV clone of the virus over the years and this has allowed us to carry out both in vitro and in vivo studies of MMTV replication and tumorigenesis using the same viral strain and mouse genetic background.  This has resulted in reducing two important confounding factors that could have affected our results or made it difficult to compare results from different studies.  This is especially true of gene expression studies since ultimately, we would like to conduct this study using HYBMTV-infected mammary glands and compare them to uninfected mammary glands and mammary tumors from the same litter mates of the BALB/c mice.  Therefore, having a BALB/c background and the same viral strain will allow us to compare this future study with our current one conducted in mammary epithelial cells only.

4)    The discussion on the comparison with transgenic mouse models should be weakened since these are somewhat artificial as compared to the real life situation. A particular concern is why the authors focus on the PyMT mouse model which uses mammary targeted expression of an oncogene that has not been shown to be involved in mammary tumorigenesis and why not other transgenic mouse models where oncogenes that have been implicated in breast cancer are targeted to the mammary gland such as MMTV-Neu and MMTV-H-Ras. 

Response: It is interesting how different reviewers see the importance of different aspects of the manuscript.  It is upon a particular reviewer’s request that we had added the PyMT mouse model even though, like the present reviewer, we felt that this model was not really relevant to MMTV biology.  To compensate for that lacunae, on our own impetus, we added the Wnt1 model being more appropriate to MMTV life cycle.  In both models, the MMTV LTR has been used to target the specific oncogene to the mammary gland to induce mammary tumors.  When we did the analysis, we actually found several intriguing parallels and similarities between the data, aspects we have already discussed in the original manuscript.  We agree with the reviewer that our assertions are somewhat speculative; however, one purpose of this study was to open up newer perspectives on how MMTV affects mammary epithelial cell gene expression, leading to its transformation.  Thus, the added discussion brings forth these ideas, such as how MMTV may be exploiting methylation as an epigenetic switch to facilitate its replication and passage to the next generation, or support the existing hypothesis that one or more of its  genes (such as env or gag) may have oncogenic potential to induce cell transformation other than the current model of “insertional mutagenesis”. 

In addition to the two mouse models, we compared our data to changes in gene expression observed in human breast cancer as well since, as mentioned by the reviewer, MMTV may be jumping the species barrier.  Now, we could add more models, such as MMTV-Neu or MMTV-H-Ras, but unfortunately for one, RNAseq data for these models does not exist to the best of our knowledge.  But even if that data was available, we believe it would not have enhanced the manuscript beyond what it currently offers since it would all still be speculative.

5)    The authors should discuss their data in light of the study by Thrasyvoulou et al., 2020 (VL30 retrotransposition is associated with induced EMT, CSC generation and tumorigenesis in HC11 mouse mammary stem‐like epithelial cells. Oncology Reports 44: 126–138). 

Response: As suggested by the respected reviewer, we have discussed our data in light of the study by Thrasyvoulou et al., 2020 (see lines 737-742 of the revised text.

6)    In general, the discussion is overly long and contains much speculation.

Response: We agree with the respected reviewer and as suggested, we have shortened and streamlined the Discussion section.  Briefly, to achieve this properly without drastically changing the current text too much (which has already been approved by other reviewers with no objection), we have moved the discussion on the hub genes to the supplementary data since that forms as independent unit not affecting other parts of the manuscript and the readers can easily go to the supplementary material (Supplementary File S1) to get more information about these hub genes, if interested.  This has allowed us to shorten the discussion, while maintaining the focus of the manuscript, bring in the new perspectives into the exploration of MMTV replication and virus-induced tumorigenesis, and yet not change the manuscript too much.  We hope that the reviewer will agree with this balanced approach and at the same time, not take away too much from the manuscript since a lot of effort was put in to add the comparison with the mouse models which may be of interest to many readers.

Minor Point

Lines 63 “Most of the endogenous strains of Mtvs are observed to be defective..”

The authors could reference Salmons and Günzburg (1987) Current Perspectives in the Biology of Mouse Mammary Tumour Virus. Virus Research 8: 81-102

Response: The suggested, reference (no 14) has been added to the revised manuscript on line 65.

Typo

We also observed induced pathway in our study in MMTV phenotype although it was not statistically significant It has been evident that viral infection dysregulates immune system and increased expression of Icam-1 and other adhesion molecules like integrins and angiogenic proteins

Response: Thank you for pointing this error.  As mentioned, it was a typo that has been corrected in the revised manuscript.

Round 2

Reviewer 1 Report

I approve the revision.

Reviewer 3 Report

The authors have adequately addressed most of the comments I had.